# Learning Causal Effects via Weighted Empirical Risk Minimization

**Yonghan Jung**
Department of Computer Science
Purdue University University
West Lafayette, IN 47907
jung222@purdue.edu

**Jin Tian**
Department of Computer Science
Iowa State University
Ames, IA 50011
jtian@iastate.edu

**Elias Bareinboim**
Department of Computer Science
Columbia University
New York, NY 10027
eb@cs.columbia.edu

## Abstract

Learning causal effects from data is a fundamental problem across the sciences. Determining the identifiability of a target effect from a combination of the observational distribution and the causal graph underlying a phenomenon is well-understood in theory. However, in practice, it remains a challenge to apply the identification theory to estimate the identified causal functionals from finite samples. Although a plethora of effective estimators have been developed under the setting known as the *back-door* (also called *conditional ignorability*), there exists still no systematic way of estimating arbitrary causal functionals that are both computationally and statistically attractive. This paper aims to bridge this gap, from causal identification to causal estimation. We note that estimating functionals from limited samples based on the *empirical risk minimization* (ERM) principle has been pervasive in the machine learning literature, and these methods have been extended to causal inference under the back-door setting. In this paper, we develop a learning framework that marries two families of methods, benefiting from the generality of the causal identification theory and the effectiveness of the estimators produced based on the principle of ERM. Specifically, we develop a sound and complete algorithm that generates causal functionals in the form of weighted distributions that are amenable to the ERM optimization. We then provide a practical procedure for learning causal effects from finite samples and a causal graph. Finally, experimental results support the effectiveness of our approach.

## 1 Introduction

Inferring causal effects from data is a fundamental challenge that cuts across the empirical sciences [35, 47, 36]. There exists a growing literature trying to delineate the conditions under which causal conclusions can be drawn from non-experimental data. One common task in the field is known as the problem of *causal effect identification* (identification, for short). Identification asks whether a causal distribution $P(Y = y | do(X = x))$ (for short, $P(y|do(x))$) can be uniquely computed from a combination of the observational distribution $P(V)$ and qualitative knowledge about the domain, which is usually encoded as a causal graph $G$ [35, Def. 3.2.4]. Causal identification has been

extensively studied based on the *do-calculus* [34], and complete graphical and algorithmic conditions have been developed for variants of this problem [51, 22, 46, 2, 3, 24, 31, 30].

For concreteness, consider the task of identifying the effect of $X$ on $Y$, $P(y|do(x))$, from the causal graph $G$ in Fig. 1a and an observational distribution $P(v)$, where $V = \{Z, X, Y\}$ is the set of observed variables. An identification algorithm will return an expression such as $P(y|do(x)) = \sum_z P(y|x, z)P(z)$. In words, this means that the target effect (l.h.s.), which is unobserved, is equal to a function of observed quantity, shown on the r.h.s.. Remarkably, the result of this analysis – the identification expression – is given in terms of distributions, and one needs to go further and estimate this quantity from finite samples, providing a realizable, empirical estimator of the r.h.s.. In practice, estimating arbitrary causal expressions from finite samples is challenging, both statistically and computationally. The only viable general-purpose method is the "plug-in estimator" [10], which estimates conditional probabilities (e.g., $P(y|z, x)$) by imposing parametric model assumptions. However, the method suffers computationally on high dimensional data [15].

One prominent setting where effective estimators have been developed is when the *back-door (BD)* condition holds [35, Sec. 3.3.1] (known as *ignorability* in statistics [43]), as in Fig. 1(a). In fact, there exist a plethora of efficient and computationally attractive statistical estimators to evaluate the BD functionals, also known as *g-formula* [40], including [42, 37, 41, 1, 52, 21], just to cite a few. More recently, different estimators were developed for identifiable effects in a few settings that go beyond the BD [27, 16, 6]. Despite all the power achieved by these methods, there is a gap from causal effect "identification" to "estimation" in that there exists still no systematic way of estimating *arbitrary* identifiable functionals that are both computationally and statistically attractive.

On a different thread, the challenge of estimating functionals has been pervasive throughout the machine learning literature, which is especially acute in higher dimensions. The issue of generalizability from a sample to the corresponding population is often studied through the principle of *structural risk minimization* [53]. This principle has been applied successfully across a number of applications [29, 5, 7, 19, 13, 32, 56, 14]. In domain adaptation, for instance, the issue of generalization is salient, and one would train a *weighted* predictor on a target domain using data from a source domain by employing what is known as the *weighted empirical risk minimization* (WERM) method [45, 5, 18, 39, 4, 12, 57, 9, 55]. These results have been extended and applied to causal inference settings as well, where the generalization step is from the observational to the experimental domain [8, 48, 25, 44, 26, 33, 20]. For instance, one could take an WERM approach leveraging weights (such as the ones coming from the inverse-probability weighting (IPW)) computed in the observational domain to answer an experimental query in the target [48, 26]. However, the prior work on applying WERM to causal inference is limited to settings contingent on the BD/ignorability assumption.

The goal of this paper is to develop a learning framework that could work for *any* identifiable causal functional without the BD/ignorability assumption, by marrying two families of methods, benefiting from the generality of the causal identification methods based on graphs (i.e., ID) and the effectiveness of the estimators produced based on the principle of WERM. We call the proposed new framework by *WERM-ID*, shortcut for *Weighted Empirical Risk Minimization for Causally Identifiable Functionals*. We exemplify the difficulties of this marriage for both spouses (WERM and ID) through a simple example.

**Example 1.** The causal graph in Fig. 1b illustrates a data-generating process of an observational study that leverages a surrogate endpoint $X$, a variable intended to substitute for a clinical endpoint $Y$ when the clinical endpoint is hardly accessible [27]. Suppose one is interested in estimating the causal effect of CD4 cell counts ($X$) on Progression of HIV ($Y$) to support the use of CD4 cell counts as a surrogate endpoint [23]. $R$ denotes the treatment that affects the CD4 cell counts, $W$ is a set of confounders affecting the treatment (e.g., the previous disease history), and there are unmeasured confounders (represented by the dashed bidirected arrows) between $W$ and $X$, and $W$ and $Y$, respectively . If one runs a standard identification algorithm [46],

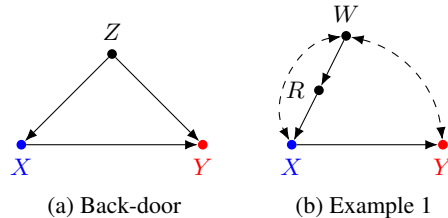

(a) Back-door  (b) Example 1

Figure 1: Causal graphs corresponding to BD and Example 1. Nodes representing the treatment and outcome are colored in blue and red, respectively.

the resultant estimand is given by:[1]

$$P\left(y|do(x)\right) = (\sum_w P(y,x|r,w)P(w))/(\sum_w P(x|r,w)P(w)). \tag{1}$$

The target effect $P\left(y|do(x)\right)$ (l.h.s.) can be estimated through the ratio of these two quantities (r.h.s.), which is not in the form of a BD expression. Unfortunately, a standard WERM solver cannot take this expression as input since it doesn't conform with the expected form. In other words, the output of a standard ID algorithm cannot be used as the input for a standard WERM procedure. $\qquad\square$

One may be tempted to surmise that the marriage may not be viable, after all, these two are perhaps qualitatively different species. Perhaps surprisingly, as formally shown in Section 3, this is just a small bump in the relationship, which can be circumvented by writing the causal estimand Eq. (1) in a friendlier form. Specifically, if we define a weighted distribution in the form of $P^{\mathcal{W}}(x,y,w,r) = \mathcal{W}P(x,y,w,r)$ with a weight function $\mathcal{W} = P(r)/P(r|w)$, then the r.h.s. of Eq. (1) can be rewritten as $P^{\mathcal{W}}(y|x,r)$. As a consequence, on the lens of WERM, the causal effect $P(y|do(x)) = P^{\mathcal{W}}(y|x,r)$ could be realized by learning the conditional distribution $P(y|x,r)$ using the samples of the original $P$ weighted by $\mathcal{W}$. This demonstrates an instance of the WERM-ID marriage.

This observation leads to the question on whether other identifiable functionals can be converted to a form that is amenable to WERM optimization. In this paper, we answer to this question in full generality, and provide a learning framework that combines the theory of causal effect identification with the principle of WERM and works for any identifiable causal functional without the BD/ignorability assumption. More specifically, our contributions are as follows:

1. We develop a sound and complete algorithm that generates *any* identifiable causal functionals as weighted distributions, amenable to WERM method.

2. We formulate the causal estimation problem as an WERM optimization. We introduce a learning objective, inspired by generalization error bound, and provide theoretical learning guarantee to the solution.

3. We develop a practical and systematic algorithm for learning target causal effects from finite samples given a causal graph, based on the proposed framework. The practical effectiveness of this approach is demonstrated through simulated studies.

Due to space constraints, the proofs are provided in the [28, Appendix C].

## 2 Preliminaries

**Structural Causal Models.** We use the language of structural causal models (SCMs) [35, pp. 204-207] as our basic semantical framework. Each SCM $\mathcal{M}$ over a set of variables $\mathbf{V}$ induces a distribution $P_{\mathcal{M}}(\mathbf{v})$ (shortly, $P(\mathbf{v})$) and has a causal graph $G$ associated to it in which solid-directed arrows encode functional relationships between observed variables, and dashed-bidirected arrows encode unobserved common causes (e.g., see Fig. 1b). Within the structural semantics, performing an intervention, and setting $\mathbf{X} = \mathbf{x}$, is represented through the do-operator, $do(\mathbf{X} = \mathbf{x})$, which encodes the operation of replacing the original equations of $\mathbf{X}$ by the constant $\mathbf{x}$, and induces a submodel $M_{\mathbf{x}}$ and an experimental distribution $P\left(\mathbf{v}|do(\mathbf{x})\right)$. For a detailed discussion of SCMs, refer to [35].

**Notations.** Each variable will be represented with a capital letter $(X)$ and its realized value with the small letter $(x)$. We will use bold letters $(\mathbf{X})$ to denote sets of variables. Given an ordered set of variables $\mathbf{X} : X_1 < \cdots < X_n$, we denote $\mathbf{X}^{(i)} = \{X_1, \cdots, X_i\}$. We use $An(\mathbf{C})_G$ to represent the union of $\mathbf{C}$ with its ancestors in the graph $G$. $G_{\mathbf{C}}$ denotes the subgraph of $G$ over $\mathbf{C}$. $\mathbb{E}_P[f(\mathbf{Y})|\mathbf{x}]$ denotes the conditional expectation of $f(\mathbf{Y})$ over $P(\mathbf{y}|\mathbf{x})$. We will adopt weighting based techniques for estimating causal effects, utilizing the following notation:

**Definition 1 (Weighted distribution $P^{\mathcal{W}}(\mathbf{v})$).** Given a distribution $P(\mathbf{v})$ and a weight function $0 < \mathcal{W}(\mathbf{v}) < \infty$ such that $\mathbb{E}_P[\mathcal{W}(\mathbf{V})] = 1$ and $\mathbb{E}_P[\mathcal{W}^2(\mathbf{V})] < \infty$, a weighted distribution $P^{\mathcal{W}}(\mathbf{v})$ is given by $P^{\mathcal{W}}(\mathbf{v}) \equiv \mathcal{W}(\mathbf{v})P(\mathbf{v})$.

**Algorithm 1:** wID $(\mathbf{x}, \mathbf{y}, G, P)$

---

**Input:** $\mathbf{x}, \mathbf{y}, G, P$
**Output:** Expression of $P(\mathbf{y}|do(\mathbf{x}))$ as a weighted distribution; or FAIL if $P(\mathbf{y}|do(\mathbf{x}))$ is unidentifiable.

1 Let $\mathbf{V} \leftarrow An(\mathbf{Y})$; $P(\mathbf{v}) \leftarrow P(An(\mathbf{Y}))$; and $G \leftarrow G_{An(\mathbf{Y})}$.
2 Find the $C$-components of $G$: $\mathbf{S}_1, \cdots, \mathbf{S}_k$.
3 Let $Q[\mathbf{S}_i] = P^{\mathcal{W}_{\mathbf{s}_i}}(\mathbf{s}_i|\mathbf{r}_{\mathbf{s}_i})$ where $(\mathcal{W}_{\mathbf{s}_i}, \mathbf{r}_{\mathbf{s}_i})$ are derived from Lemma 1.
4 Let $\mathbf{D} \equiv An(\mathbf{Y})_{G_{\mathbf{V}\setminus\mathbf{x}}}$.
5 Find the $C$-component of $G_{\mathbf{D}}$: $\mathbf{D}_1, \cdots \mathbf{D}_K$.
6 For each $\mathbf{D}_i \in \mathbf{S}_j$ for some $(i, j)$, let
   $\quad Q[\mathbf{D}_i] = \texttt{wIdentify}\left(\mathbf{D}_i, \mathbf{S}_j, Q[\mathbf{S}_j], \mathbf{r}_{\mathbf{s}_j}, \mathcal{W}_{\mathbf{s}_j}\right) \equiv P^{\mathcal{W}_{\mathbf{d}_i}}(\mathbf{d}_i|\mathbf{r}_{\mathbf{d}_i})$.
7 **if** $K = 1$ **then**
   |    **return** $P(\mathbf{y}|do(\mathbf{x})) = P^{\mathcal{W}_{\mathbf{d}_1}}(\mathbf{y}|\mathbf{r}_{\mathbf{d}_1})$.
   **end**
8 Let $\mathcal{W} \equiv \prod_{i=1}^{K} P^{\mathcal{W}_{\mathbf{d}_i}}(\mathbf{d}_i|\mathbf{r}_{\mathbf{d}_i}) / P(\mathbf{d}|\mathbf{r})$ where $\mathbf{R} \equiv \mathbf{V}\setminus\mathbf{D}$.
9 **return** $P(\mathbf{y}|do(\mathbf{x})) = P^{\mathcal{W}}(\mathbf{y}|\mathbf{r})$
  **Procedure** wIdentify$(\mathbf{C}, \mathbf{T}, Q[\mathbf{T}], \mathbf{r}, \mathcal{W})$
    | **Input:** $\mathbf{T}, Q[\mathbf{T}] = P^{\mathcal{W}}(\mathbf{t}|\mathbf{r})$
    | **Output:** $Q[\mathbf{C}]$ for $\mathbf{C} \subseteq \mathbf{T}$ as a weighted distribution.
a.1   | Let $\mathbf{A} \equiv An(\mathbf{C})_{G_{\mathbf{T}}}$, then $Q[\mathbf{A}] = P^{\mathcal{W}}(\mathbf{a}|\mathbf{r})$ by Lemma 1.
a.2   | **if** $\mathbf{A} = \mathbf{C}$ **then**
    |    **return** $Q[\mathbf{C}] = P^{\mathcal{W}}(\mathbf{a}|\mathbf{r})$
    | **end**
a.3   | **if** $\mathbf{A} = \mathbf{T}$ **then**
    |    **return** *FAIL*
    | **end**
a.4   | **else**
a.5     | Let $\mathbf{S}$ denote the $C$-component in $G_{\mathbf{A}}$ such that $\mathbf{C} \subseteq \mathbf{S}$.
a.6     | Compute $Q[\mathbf{S}] = P^{\mathcal{W}\times\mathcal{W}'}(\mathbf{s}|\mathbf{r}')$ where $(\mathcal{W}', \mathbf{r}')$ are derived by Lemma 1.
a.7     | **return** wIdentify$(\mathbf{C}, \mathbf{S}, Q[\mathbf{S}], \mathbf{r}', \mathcal{W}\times\mathcal{W}')$
    | **end**

---

**Causal Effects Identification** Given a causal graph $G$ over a set of variables $\mathbf{V}$, a causal effect $P(\mathbf{y}|do(\mathbf{x}))$ is said to be *identifiable* from $G$ if $P(\mathbf{y}|do(\mathbf{x}))$ is uniquely computable from the observed distribution $P(\mathbf{v})$ in any SCM that induces $G$ [35, pp.77]. Complete causal effects identification algorithms have been developed using a decomposition strategy of the causal graph based on *confounded components*:

**Definition 2** ($C$-**component [49]**). In a causal graph $G$, two variables are said to be in the same confounded component (for short, $C$-component) if and only if they are connected by a bi-directed path, i.e., a path composed solely of bi-directed edges $V_i \leftrightarrow V_j$.

For any $\mathbf{C} \subseteq \mathbf{V}$, the quantity $Q[\mathbf{C}]$, called $C$-factor, is defined as the post-intervention distribution of $\mathbf{C}$ under an intervention on $\mathbf{V}\setminus\mathbf{C}$: $Q[\mathbf{C}] \equiv P(\mathbf{c}|do(\mathbf{v}\setminus\mathbf{c}))$. [51] showed that the joint distribution $P(\mathbf{v})$ is factorized by $C$-factors as $P(\mathbf{v}) = \prod_i Q[\mathbf{S}_i]$, where $\mathbf{S}_i$ are the set of $C$-components of $G$, and developed a complete causal identification algorithm based on the decomposition of $C$-factors.

## 3 Representing Causal Effects as Weighted Distributions

In this section, we present a sound and complete algorithm that expresses *any* identifiable causal effects in the form of weighted distributions. This result allows the use of WERM method for estimating causal effects from finite samples presented in Section 4.

By and large, we note that the functionals returned by a standard identification algorithm for a target causal effect (whenever identifiable) are not in the form of weighted distributions that are amenable to WERM approach. To witness that this is possible, we analyze the causal functional given in Eq. (1), the resultant of an identification algorithm, and manually convert it into a weighted form. First note that the numerator/denominator in Eq. (1) are both in the form of a BD adjustment, such that the numerator can be rewritten as the effect of $R$ on $\{X, Y\}$: $P(x, y|do(r)) = \sum_w P(y, x|r, w)P(w)$, and the denominator the effect of $R$ on $\{X\}$: $P(x|do(r)) = \sum_w P(x|r, w)P(w)$, both with $W$ as the adjustment set. Therefore the causal effect in Eq. (1) can be written as

$$P(y|do(x)) = P(x, y|do(r))/P(x|do(r)).\tag{2}$$

We then use a well-known result for the BD adjustment that the effect of $R$ on $\{X, Y\}$ with $W$ as the adjustment set can be written as a weighted distribution $P(x, y | do(r)) = P^{\mathcal{W}}(x, y | r)$ where the stabilized IPW weight is given by $\mathcal{W} = P(r)/P(r|w)$. We obtain that the causal effect in Eq. (2) can be written as $P(y|do(x)) = P(x, y | do(r)) / P(x | do(r)) = P^{\mathcal{W}}(y|x, r)$. This weighted distribution form allows the causal effect to be estimated via using WERM to learn conditional distributions from weighted samples.

Rather than having to manually transform a causal functional into a weighted distribution form, our goal is to develop a fully systematical algorithm that can express any identifiable causal effects in the form of weighted distributions. We develop our algorithm based on the identification algorithm in [51] which main idea was to identify $C$-factors through recursively marginalizing and decomposing $C$-components. We prove next a key result that rewrites the marginalization and $C$-components decomposition operations in terms of weighted distributions:

**Lemma 1** (**Computing C-factors as weighted distributions**). *Let a topological order over* $\mathbf{V}$ *be* $V_1 < V_2 < \cdots < V_n$. *Suppose* $Q[\mathbf{A}]$ *is given by* $Q[\mathbf{A}] = P^{\mathcal{W}}(\mathbf{a}|\mathbf{r})$ *for some* $\mathbf{R} \subseteq \mathbf{V}$ *and weight function* $\mathcal{W}$.

*1. If* $\mathbf{W}$ *is a* $C$-*component of* $G_{\mathbf{A}}$, *then* $Q[\mathbf{W}] = P^{\mathcal{W} \times \mathcal{W}'}(\mathbf{w}|\mathbf{r}')$, *where* $\mathbf{R}' \equiv \mathbf{R} \cup ((\mathbf{A} \backslash \mathbf{W}) \cap An(\mathbf{W}))$ *and* $\mathcal{W}' \equiv \frac{P^{\mathcal{W}}((\mathbf{a}\backslash\mathbf{w}) \cap An(\mathbf{w})|\mathbf{r})}{\prod_{V_i \in (\mathbf{A}\backslash\mathbf{W}) \cap An(\mathbf{W})} P^{\mathcal{W}}\left(v_i | \mathbf{v}^{(i-1)} \cap \mathbf{a} \cap An(\mathbf{w}), \mathbf{r}\right)}$.

*2. If* $\mathbf{W} \subseteq \mathbf{A}$ *satisfies* $\mathbf{W} = An(\mathbf{W})_{G_{\mathbf{A}}}$, *then* $Q[\mathbf{W}] = P^{\mathcal{W}}(\mathbf{w}|\mathbf{r})$.

In words, the proposition will recursively compute $C$-factors in terms of weighted distributions. The base case is $\mathbf{A} = \mathbf{V}$, and we have $Q[\mathbf{V}] = P(\mathbf{v}) = P^{\mathcal{W}_0}(\mathbf{v}|\mathbf{r}_0)$ with $\mathcal{W}_0 = 1$ and $\mathbf{R}_0 = \emptyset$. If $\mathbf{S}_i$ is a $C$-component of $G$, then $Q[\mathbf{S}_i] = P^{\mathcal{W}'}(\mathbf{s}_i|\mathbf{r}')$ with $\mathbf{R}' = (\mathbf{V}\backslash\mathbf{S}_i) \cap An(\mathbf{S}_i)$ and $\mathcal{W}' = \frac{P(\mathbf{r}')}{\prod_{V_i \in \mathbf{R}'} P(v_i|\mathbf{v}^{(i-1)} \cap An(\mathbf{S}_i))}$. For example, the graph in Fig. 1b has two $C$-components $\mathbf{S}_1 = \{W, X, Y\}$ and $\mathbf{S}_2 = \{R\}$. By Lemma 1, we obtain $Q[\mathbf{S}_1] = P^{\mathcal{W}_1}(\mathbf{s}_1|r)$ (with $\mathbf{R}' = R$) where $\mathcal{W}_1 = P(r)/P(r|w)$, and $Q[\mathbf{S}_2] = P(r|w)$ (with $\mathbf{R}' = W$). The significance of Lemma 1 stems from the fact that it will allow one to rewrite the complete algorithm in [51] to express identifiable causal effects in the form of weighted distributions. The proposed new algorithm is shown in Algo. 1 and we have proved its equivalence with the original as follows:

**Theorem 1** (**Soundness and Completeness of Algo. 1**). *A causal effect* $P(\mathbf{y}|do(\mathbf{x}))$ *is identifiable if and only if* $\texttt{wID}(\mathbf{x}, \mathbf{y}, G, P)$ *(Algo. 1) returns* $P^{\mathcal{W}}(\mathbf{y}|\mathbf{r})$ *such that* $P(\mathbf{y}|do(\mathbf{x})) = P^{\mathcal{W}}(\mathbf{y}|\mathbf{r})$.

In words, re-writing an identifiable effect in terms of weighting entails no loss of information. This may be surprising since there is no reason to believe *a priori* that arbitrary estimands could be written in the form of weighted distributions. For concreteness, we demonstrate the application of Algo. 1 using the model in Fig. 1b, where $P(y|do(x))$ is identified as given in Eq. (1) (i.e., not in the weighting-form). The graph has two $C$-components $\mathbf{S}_1 = \{W, X, Y\}$ and $\mathbf{S}_2 = \{R\}$ (Line 2). We have $Q[\mathbf{S}_1] = P^{\mathcal{W}_1}(\mathbf{s}_1|r)$ where $\mathcal{W}_1 = P(r)/P(r|w)$, and $Q[\mathbf{S}_2] = P(r|w)$ by Lemma 1 as discussed previously (Line 3). Let $\mathbf{D} = An(Y)_{G_{\mathbf{V}\backslash x}} = \{Y\}$ (Line 4). Run $\texttt{wIdentify}(Y, \mathbf{S}_1, Q[\mathbf{S}_1], r, \mathcal{W}_1)$ (Line 6). In Procedure $\texttt{wIdentify}()$, let $\mathbf{A} = An(Y)_{G_{\mathbf{S}_1}} = \{X, Y\}$, then $Q[\mathbf{A}] = P^{\mathcal{W}_1}(\mathbf{a}|r)$ (Line a.1). In $G_{\mathbf{A}} = G_{\{X,Y\}}$, let $\mathbf{S} \equiv \{Y\}$ denote the $C$-component containing $Y$ (Line a.5). Then, $Q[\mathbf{S}] = Q[Y] = P^{\mathcal{W}_1 \times \mathcal{W}'}(y|\mathbf{r}')$ where $\mathbf{R}' = \{R, X\}$ and $\mathcal{W}' = P^{\mathcal{W}}(x|r)/P^{\mathcal{W}}(x|r) = 1$ by Lemma 1 (with $\mathbf{W} = \mathbf{S} = Y$) (Line a.6). Line a.7 returns $Q[Y] = \texttt{wIdentify}(Y, \mathbf{S}, Q[\mathbf{S}], r', \mathcal{W}_1) = P^{\mathcal{W}_1}(y|x, r)$. Finally we obtain $P(y|do(x)) = P^{\mathcal{W}_1}(y|x, r)$ (Line 7).

The importance of Thm. 1 lies in that it facilitates an end-to-end solution to causal effect estimation from finite samples: Causal graph $\rightarrow$ Determine the identifiability $\rightarrow$ Produce a causal estimand $\rightarrow$ Formulate WERM learning objective with learning guarantee $\rightarrow$ Solve the optimization problem $\rightarrow$ Estimation. We'll discuss the last steps of this pipeline in the next section.

## 4 Learning Causal Effects via Weighted Empirical Risk Minimization

Algo. 1 allows us to write any causal effect as a weighted distribution, namely, $P(\mathbf{y}|do(\mathbf{x})) = P^{\mathcal{W}^*}(\mathbf{y}|\mathbf{r})$ for some weight function $\mathcal{W}^*(\mathbf{v})$ and $\mathbf{R} \subseteq \mathbf{V}$. For example, for the model given in Fig. 1b, we have shown that $P(y|do(x)) = P^{\mathcal{W}^*}(y|r, x)$ where $\mathcal{W}^* = P(r)/P(r|w)$. In this section,

we will develop an algorithm for learning $P^{\mathcal{W}^*}(y|\mathbf{r})$, i.e., the causal effect that has been cast as a weighted distribution, using finite samples $\mathcal{D} = \{\mathbf{V}_{(i)}\}_{i=1}^m$ drawn from $P(\mathbf{v})$. We will focus on learning $E[Y|do(\mathbf{x})] = \mathbb{E}_{P^{\mathcal{W}^*}}[Y|\mathbf{r}]$.

## 4.1 Learning Setup: Weighted Empirical Risk Minimization

In the WERM setting, we attempt to learn a function $h(\mathbf{r})$ that approximates $\mathbb{E}_{P^{\mathcal{W}^*}}[Y|\mathbf{r}]$, and the task can be viewed as a supervised learning problem of choosing the best hypothesis $h(\mathbf{r})$ from a hypothesis class $\mathcal{H}$ that minimizes a loss function $\ell(h(\mathbf{r}), y)$. Here $\ell(y', y)$ is a loss function suitable for the application, for example, the squared loss $\ell(y', y) = (y' - y)^2$. Formally, the learning task can be expressed as minimizing the expected loss on $P^{\mathcal{W}^*}$, known as the *weighted risk*:

$$R^{\mathcal{W}^*}(h) \equiv \mathbb{E}_{P^{\mathcal{W}^*}}[\ell(h(\mathbf{R}), Y)] = \mathbb{E}_P\left[\mathcal{W}^*(\mathbf{V})\ell(h(\mathbf{R}), Y)\right]. \tag{3}$$

Given data $\mathcal{D} = \{\mathbf{V}_{(i)}\}_{i=1}^m$ drawn from $P(\mathbf{v})$, the corresponding *weighted empirical risk (WER)* is

$$\widehat{R}^{\mathcal{W}^*}(h) \equiv \frac{1}{m}\sum_{i=1}^m \mathcal{W}^*(\mathbf{V}_{(i)})\ell\left(h(\mathbf{R}_{(i)}), Y_{(i)}\right). \tag{4}$$

**Generalization Bound.** While minimizing the WER $\widehat{R}^{\mathcal{W}^*}(h)$ is consistent, the corresponding estimator could suffer from high variance in small samples and lead to unstable estimates [48, 26]. For instance, for $\mathcal{W}^* = P(r)/P(r|w)$, $\widehat{R}^{\mathcal{W}^*}(h)$ could have large variance if $P(r|w)$ is very small, potentially resulting in the minimizer $h$ of Eq. (4) to overfit the data. One technique to mitigate this issue is to introduce a new weight function $\mathcal{W}$ intended to be an approximation of $\mathcal{W}^*$ but with a lower variance [48, 26], leading to the re-weighted empirical risk $\widehat{R}^{\mathcal{W}}(h) \equiv \frac{1}{m}\sum_{i=1}^m \mathcal{W}(\mathbf{V}_{(i)})\ell\left(h(\mathbf{R}_{(i)}), Y_{(i)}\right)$. The relation between $\widehat{R}^{\mathcal{W}}(h)$ and the target WER $R^{\mathcal{W}^*}(h)$ is given by the following generalization error bound:

**Proposition 1 (Generalization bound [12, Thm.4]).** *Let $p$ denote the Pollard's pseudo-dimension[2] of loss function $\ell_h(\mathbf{v}) \equiv \ell(h(\mathbf{v}), \mathbf{y})$ and $\hat{P}$ denote the empirical distribution of $P$. Then, for any $\delta \in (0, 1)$, with probability at least $1 - \delta$, the following holds:*

$$|R^{\mathcal{W}^*}(h) - \widehat{R}^{\mathcal{W}}(h)| \le \underbrace{\mathbb{E}_P\left[|\mathcal{W}^*(\mathbf{V}) - \mathcal{W}(\mathbf{V})|\right]}_{(a)} + 2^{5/4}\max\underbrace{\left(\sqrt{\mathbb{E}_P\left[\mathcal{W}^2\ell_h^2\right]}, \sqrt{\mathbb{E}_{\hat{P}}\left[\mathcal{W}^2\ell_h^2\right]}\right)}_{(b)}\underbrace{F(p, m, \delta)}_{(c)},$$

$$\tag{5}$$

*where $F(p, m, \delta) \equiv \left((p\log(2me/p) + \log(4/\delta))^{3/8}\right)/(m^{3/8})$.*

Prop. 1 implies that the distance between $\mathcal{W}$ and $\mathcal{W}^*$ (Eq. (5a)), the second moment (variance) of $\mathcal{W}$ (Eq. (5b)), and the pseudo-dimension $p$ of $\ell_h$ (Eq. (5c)) all contribute to the error bound. In particular, even though directly minimizing WER $\widehat{R}^{\mathcal{W}^*}(h)$ (setting $\mathcal{W} = \mathcal{W}^*$) may lead to an estimator with small bias, the results can still suffer from high variance due to Eq. (5b).

**Learning Objective.** Motivated by Prop. 1, we propose to simultaneously learn a hypothesis $h$ that minimizes $\widehat{R}^{\mathcal{W}}(h)$ and a weight function $\mathcal{W}$ that approximate $\mathcal{W}^*$ while penalizing the variance of $\mathcal{W}$, adopting the common idea of minimizing a upper bound of the target risk [18, 48, 26, 14]. Specifically, we propose the following learning objective

$$\mathcal{L}(\mathcal{W}, h) \equiv \underbrace{\widehat{\mathcal{R}}^{\mathcal{W}}(h) + \frac{\lambda_h}{m}C(h)}_{\mathcal{L}_h(h, \mathcal{W}, \lambda_h)} + \underbrace{\sqrt{\frac{1}{m}\sum_{i=1}^m\left(\mathcal{W}(\mathbf{V}_{(i)}) - \mathcal{W}^*(\mathbf{V}_{(i)})\right)^2 + \frac{\lambda_{\mathcal{W}}}{m}\|\mathcal{W}\|_2^2}}_{\mathcal{L}_{\mathcal{W}}(\mathcal{W}, \lambda_{\mathcal{W}}; \mathcal{W}^*)}, \tag{6}$$

where $\mathcal{L}_h(h, \mathcal{W}, \lambda_h)$ consists of the WER $\widehat{R}^{\mathcal{W}}(h)$ and a regularizer $C(h)$ of $h$, such as $L_1$ or $L_2$ regularization for the parameters of $h$; $\mathcal{L}_{\mathcal{W}}(\mathcal{W}, \lambda_{\mathcal{W}}; \mathcal{W}^*)$ measures the deviance of $\mathcal{W}$ from $\mathcal{W}^*$ with $L_2$ regularization to penalize the variance of $\mathcal{W}$; and $(\lambda_h, \lambda_{\mathcal{W}})$ are hyperparameters.

The objective function proposed in Eq. (6) is validated by ensuring that its minimizers converge to the minimizer of the target risk, as shown by the following result:

**Theorem 2 (Learning guarantee).** *Let* $h^* \equiv \arg\min_{h \in \mathcal{H}} \mathcal{R}^{\mathcal{W}^*}(h)$, *and* $(\mathcal{W}_m, h_m) \equiv \arg\min_{\mathcal{W} \in \mathcal{H}_\mathcal{W}, h \in \mathcal{H}} \mathcal{L}(\mathcal{W}, h)$, *where* $\mathcal{H}_\mathcal{W}$ *is the model hypotheses class for* $\mathcal{W}$. *Suppose* $\mathcal{H}_\mathcal{W}$ *is correctly specified such that* $\mathcal{W}^* \in \mathcal{H}_\mathcal{W}$. *Then,* $h_m$ *converges to* $h^*$ *with a rate of* $O_p(m^{-1/4})$. *Specifically,* $\mathcal{R}^{\mathcal{W}^*}(h_m) - \mathcal{R}^{\mathcal{W}^*}(h^*) \leq O_p(m^{-1/4})$.

In words, the theorem ascertains that the hypothesis $h_m$ that minimizes the objective function $\mathcal{L}(\mathcal{W}, h)$ in Eq. (6) converges to the hypothesis $h^*$ that minimizes the target weighted risk $\mathcal{R}^{\mathcal{W}^*}(h)$ in the limit of infinite samples.

## 4.2 Learning Algorithm

Putting these results together, we present in this section a practical procedure for learning the causal effect $\mathbb{E}[Y|do(\mathbf{x})]$ from finite samples $\mathcal{D} = \{\mathbf{V}_{(i)}\}_{i=1}^m$ in a given causal graph $G$. The first step is to run Algo. 1 $\texttt{wID}(\mathbf{x}, \mathbf{y}, G, P)$ to derive the target estimand $\mathbb{E}[Y|do(\mathbf{x})] = \mathbb{E}_{P^{\mathcal{W}^*}}[Y|\mathbf{r}]$. Then, we compute the weight $\mathcal{W}^*$ as an input to the objective function $\mathcal{L}(\mathcal{W}, h)$ in Eq. (6). In practice, we only have access to an estimate $\widehat{\mathcal{W}^*}$ of the true $\mathcal{W}^*$. In general, $\mathcal{W}^*$ may be expressed in terms of weighted distributions, e.g., $\mathcal{W}^* \equiv \frac{P^{\mathcal{W}'}(w,y|r,z)P(z|w,x)}{P(w,z,y|r,x)}$ where $\mathcal{W}'$

---

**Algorithm 2:** $\texttt{WERM-ID-R}(\mathcal{D}, G, \mathbf{x}, y)$

**Output:** An estimate of $\mathbb{E}[Y|do(\mathbf{x})]$ from data sample $\mathcal{D}$

1 Run $\texttt{wID}(\mathbf{x}, y, G, P)$ and derive $(\mathcal{W}^*, \mathbf{R})$ such that $P(y|do(\mathbf{x})) = P^{\mathcal{W}^*}(y|\mathbf{r})$.
2 Evaluate $\widehat{\mathcal{W}^*}$ from $\mathcal{D}$.
3 Learn $\mathcal{W} \equiv \arg\min_{\mathcal{W}' \in \mathcal{H}_\mathcal{W}} \mathcal{L}_\mathcal{W}(\mathcal{W}', \lambda_\mathcal{W}; \widehat{\mathcal{W}^*})$.
4 Learn $h \equiv \arg\min_{h' \in \mathcal{H}} \mathcal{L}_h(h', \mathcal{W}, \lambda_h)$.
**return** $\widehat{\mathbb{E}}[Y|do(\mathbf{x})] \equiv h(\mathbf{r})$

---

may be expressed in other weighted distributions. We estimate $\hat{P}(z|w,x)$ from $\mathcal{D}$ using some regression functions. To estimate a weighted distribution $P^{\mathcal{W}'}(w,y|r,z)$, we propose: (1) evaluate $W'$, (2) draw samples $\mathcal{D}^{\mathcal{W}'}$ that could be treated as if they were drawn from $P^{\mathcal{W}'}(\mathbf{v})$, and (3) evaluate $P^{\mathcal{W}'}(w,y|r,z)$ using $\mathcal{D}^{\mathcal{W}'}$. This recursive procedure avoids computational cost of marginalizing high-dimensional variables. A procedure for evaluating $\widehat{\mathcal{W}^*}$ is provided in Appendix B.

Learning $\mathcal{W}$ and $h$ through minimizing the objective function $\mathcal{L}(\mathcal{W}, h)$ in Eq. (6) turns out challenging in practice. Here we propose a heuristic procedure that works well in practice. The proposed procedure learns $\mathcal{W}$ and $h$ separately, as follows: First, one learns a weight function $\mathcal{W}$ that approximates $\widehat{\mathcal{W}^*}$ but having regularized variance, i.e., $\mathcal{W} \equiv \arg\min_{\mathcal{W}' \in \mathcal{H}_\mathcal{W}} \mathcal{L}_\mathcal{W}(\mathcal{W}', \lambda_\mathcal{W}; \widehat{\mathcal{W}^*})$; Equipped with $\mathcal{W}$, one learns a hypothesis $h$ by minimizing the regularized WER, i.e., $h \equiv \arg\min_{h' \in \mathcal{H}} \mathcal{L}_h(h', \mathcal{W}, \lambda_h)$, where the objective functions $\mathcal{L}_\mathcal{W}(\mathcal{W}', \lambda_\mathcal{W}; \widehat{\mathcal{W}^*})$ and $\mathcal{L}_h(h', \mathcal{W}, \lambda_h)$ are given in Eq. (6). For concreteness, for $\mathcal{W}^* = P(r)/P(r|w)$, we can write

$$\mathcal{W} = \arg\min_{\mathcal{W}' \in \mathcal{H}_\mathcal{W}} \frac{1}{m} \sum_{i=1}^m (\mathcal{W}'(R_{(i)}, W_{(i)}) - \widehat{\mathcal{W}^*}(R_{(i)}, W_{(i)}))^2 + \frac{\lambda_w}{m} \|\mathcal{W}'\|_2^2, \qquad (7)$$

where $\mathcal{H}_\mathcal{W}$ is specified as the gradient boosting regression models [11] in the experimental study. For a binary variable $Y \in \{0, 1\}$, we could employ the cross-entropy loss function, which yields

$$h = \arg\min_{h' \in \mathcal{H}} -\frac{1}{m} \sum_{i=1}^m \mathcal{W}(\mathbf{V}_{(i)}) \cdot \left(Y_{(i)} \log\left(h'(X_{(i)}, R_{(i)})\right) + (1 - Y_{(i)}) \log\left(1 - h'(X_{(i)}, R_{(i)})\right)\right) + \frac{\lambda_h}{m}C(h')$$
$$(8)$$

where $\mathcal{H}$ is also specified as the gradient boosting functions in the experimental study.

The proposed procedure for estimating causal effects from finite samples is summarized in Algo. 2. The following result provides the time complexity of the procedure:

**Theorem 3 (Time complexity of Algo. 2).** *Let* $m \equiv |\mathcal{D}|$ *and* $n \equiv |\mathbf{V}|$. *Assume all weights satisfy* $0 < \mathcal{W} < c$ *for some constant* $c > 0$. *Let* $T_1(m)$ *denote the time complexity of estimating* $\hat{P}(v_i|\cdot)$ *from sample* $\mathcal{D} \sim P(\mathbf{v})$ *for* $V_i \in \mathbf{V}$. *Let* $K$ *denote the number of* $C$-*components in* $G_\mathbf{D}$ *(in Algo. 1). Let* $T_2(m)$ *denote the time complexity of minimizing* $\mathcal{L}_\mathcal{W}$ *and* $\mathcal{L}_h$. *Then, Algo. 2 runs in* $O(poly(n) + nK(mc + nT_1(m)) + T_2(m))$ *time, where* $O(poly(n))$ *is for running Algo. 1,* $O(nK(mc + nT_1(m)))$ *for evaluating* $\widehat{\mathcal{W}^*}$.

For concreteness, suppose we estimate $\hat{P}(v_i|\cdot)$ using the gradient boosting regression models, and we optimize $\mathcal{L}_\mathcal{W}$ and $\mathcal{L}_h$ by setting both $\mathcal{H}_\mathcal{W}$ and $\mathcal{H}$ as the class of gradient boosting regression models. Then, the time complexities $T_1(m)$ and $T_2(m)$ are both $O(m \log m)$ [11, Sec.4.1].

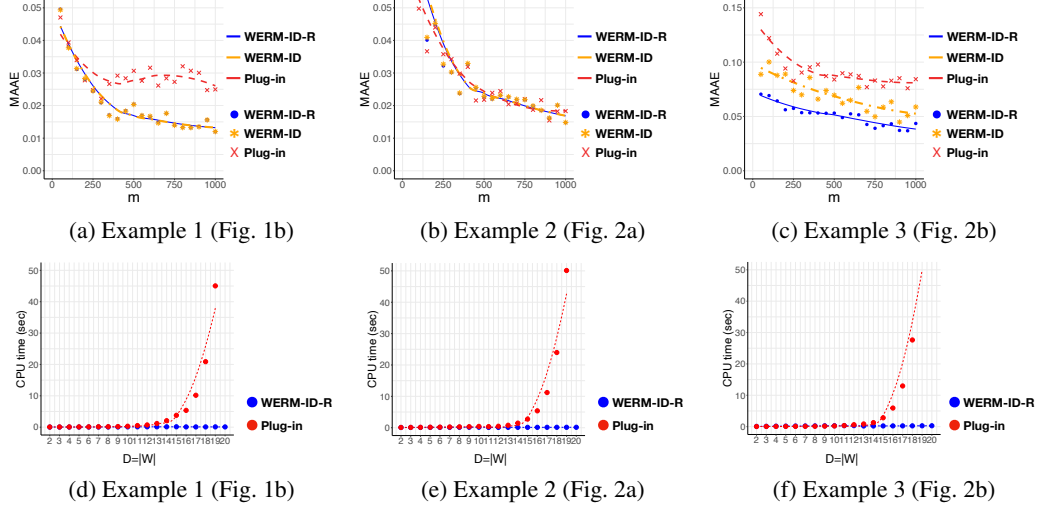

(a) Example 1 (Fig. 1b)    (b) Example 2 (Fig. 2a)    (c) Example 3 (Fig. 2b)

(d) Example 1 (Fig. 1b)    (e) Example 2 (Fig. 2a)    (f) Example 3 (Fig. 2b)

Figure 3: **(Top)** MAAE plots comparing proposed WERM based estimators (WERM-ID and WERM-ID-R) with Plug-in. **(Bottom)** Plots comparing the running time of WERM-ID-R versus Plug-in.

# 5 Experiments

We consider the following two practical examples shown in Fig. 2, in addition to Example 1. The derivation of target causal effects as weighted distributions by Algo. 1 is provided in Appendix A.

**Example 2.** In the causal graph in Fig. 2a, $X$ represents sign-up for the job-training program, $Z$ actual participation, and $Y$ the postprogram earnings [17]. Suppose there exist a set of observed confounding variables $W$ affecting $X, Y$, and $Z$, and unmeasured ones between $(W, X)$, $(X, Y)$, and $(W, Y)$ respectively. The data scientist aims to evaluate the effect of signing-up for the program on the earnings, $P(y|do(x))$. One can show that this quantity is identifiable and given by the following functional

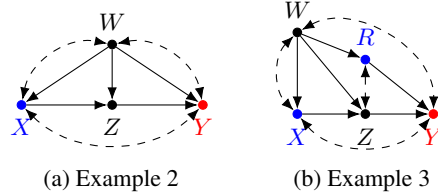

(a) Example 2    (b) Example 3

Figure 2: Causal graphs

$$P(y|do(x)) = \sum_{w,z} P(z|x,w)P(w)\sum_{x'} P(y|z,x',w)P(x'|w). \qquad (9)$$

$\square$

**Example 3.** In the causal graph in Fig. 2b, the variables represent the patients' characteristics: socioeconomic factors ($W$), diet habit ($X$), frequency of exercises ($R$), the level of cholesterol ($Z$), and the occurrence of heart-disease ($Y$). The scientist is interested in the effect of treatments $(X, R)$ on $Y$. One can show that this quantity is identifiable and given by the following functional

$$P(y|do(x,r)) = \sum_{w,z} P(z|w,x)\sum_{x'} P(y|r,w,x',z)P(x'|r,w)P(w). \qquad (10)$$

$\square$

## 5.1 Experiments Setup

We specify a SCM $M$ for each causal graph and generate datasets $\mathcal{D}$ from $M$. In order to estimate the ground truth $\mu(\mathbf{x}) \equiv \mathbb{E}[Y|do(\mathbf{x})]$, we generate $m_{int} = 10^7$ samples $\mathcal{D}_{int}$ from $M_{\mathbf{x}}$, the model induced by the intervention $do(\mathbf{X} = \mathbf{x})$, and compute the mean of $Y$ in $\mathcal{D}_{int}$.

We denote **WERM-ID-R** the estimator given in Algo. 2. $\mathcal{H}$ and $\mathcal{H}_{\mathcal{W}}$ are set as the gradient boosting regression classes. We also study a simpler variant, denoted **WERM-ID**, that directly minimizes

the WER $\widehat{R}^{\widehat{\mathcal{W}^*}}(h)$ in Eq. (4) after evaluating $\widehat{\mathcal{W}^*}$ from $\mathcal{D}$. We compare the proposed methods with the **Plug-in** estimator, the only natural method applicable to any causal functionals, which computes each conditional probability such as $P(x|r,w)$ by plugging-in the gradient boosting regression.

**Accuracy Measure.** Given a data set $\mathcal{D}$ with $m$ samples, let $\hat{\mu}_{\text{IDR}}(\mathbf{x})$, $\hat{\mu}_{\text{ID}}(\mathbf{x})$, and $\hat{\mu}_{\text{plug}}(\mathbf{x})$ be the estimated $\mathbb{E}[Y|do(\mathbf{x})]$ using the WERM-ID-R, WERM-ID, and Plug-in estimators. For each $\hat{\mu} \in \{\hat{\mu}_{\text{IDR}}, \hat{\mu}_{\text{ID}}, \hat{\mu}_{\text{plug}}\}$, we compute the average absolute error (AAE) as $|\mu(\mathbf{x}) - \hat{\mu}(\mathbf{x})|$ averaged over $\mathbf{x}$. We generate 100 datasets for each sample size $m$. We call the median of the 100 AAEs the *median average absolute error*, or MAAE, and its plot vs. the sample size $m$, the *MAAE plot*.

### 5.2 Experimental Results

We evaluate the proposed WERM learning framework against the plug-in estimators in Examples (1,2,3). All variables are binary except that $W$ *is set to be a vector of $D$ binary variables* to represent high-dimensional covariates. The detailed description of the corresponding SCMs are provided in Appendix D.

**Example 1 (Fig. 1b).** We test on estimating $\mathbb{E}[Y|do(x)]$ with $D = 15$ where the causal effect $P(y|do(x))$ is given by Eq. (1). The MAAE plots are given in Fig. 3a. We observe that the WERM-based methods (WERM-ID/WERM-ID-R) significantly outperform Plug-in.

**Example 2 (Fig. 2a).** We test on estimating $\mathbb{E}[Y|do(x)]$ with $D = 15$ where the effect $P(y|do(x))$ is given by Eq. (9). The MAAE plots are given in Fig. 3b. We observe that the WERM-based methods (WERM-ID/WERM-ID-R) perform on par with Plug-in.

**Example 3 (Fig. 2b).** We test on estimating $\mathbb{E}[Y|do(x,r)]$ with $D = 15$ where $P(y|do(x,r))$ is given by Eq. (10). The MAAE plots are given in Fig. 3c. We note that WERM-ID-R significantly outperforms WERM-ID, and both significantly outperform Plug-in.

**CPU Run Time.** We show the CPU run time plots of WERM-ID-R versus Plug-in over increasing $D$ in Fig. 3d, 3e, 3f. The run time plots of WERM-ID always overlap with WERM-ID-R and are therefore not shown. For each given $D$, we collect 100 run times and plot the median. We note that, in all three experiments, the run time of Plug-in increases rapidly over $D$ (due to the marginalization over $W$), while WERM-ID/WERM-ID-R scales well.

The reason for choosing the possibly naive, plug-in estimator as the baseline for comparison is because it constitutes the only viable estimator known to date for arbitrary identifiable functionals. Specifically, for Examples 2 and 3, we are not aware of any applicable estimators in the literature, and the only applicable one for Example 1 is CWO in [27], which is the same as WERM-ID for this example. The existing regression or weighting based methods are only applicable when the BD/ignorability assumption holds (e.g., Fig. 1a). In this case, the proposed WERM-ID estimator reduces to the standard weighted regression estimator.s More experimental results over the three examples with varying $D$ are given in Appendix D. The results consistently show that the accuracy of WERM-based estimators are never worse, and mostly superior, against the plug-in estimators. Also, WERM-ID-R usually performs better than WERM-ID.

## 6 Conclusion

This paper aims to fill the gap from causal identification to causal estimation. To this end, we developed a learning framework that brings together the causal identification theory and powerful ERM methods. In particular, we derived a sound and complete algorithm that produces causal functionals in the form of weighted distributions that are amenable to ERM optimization. We proposed a learning objective based on the WERM theory and provided a practical learning algorithm for estimating causal effects from finite samples. The effectiveness of the proposed methods had been corroborated with experimental studies. We hope that the conceptual framework and practical methods introduced in this work can inspire future investigation in the ML and CI communities towards the development of robust and efficient methods for learning causal effects in applied settings.

### Acknowledgement

We thank the reviewers for their valuable feedback helping improving the paper. Elias Bareinboim and Yonghan Jung were partially supported by grants from NSF IIS-1704352 and IIS-1750807 (CAREER). Jin Tian was partially supported by NSF grant IIS-1704352 and ONR grant N000141712140.

## Broader Impact

Learning causal effects is essential throughout the data-driven sciences. Notable merits of our work include providing a practical solution to the estimation of causal effects from finite samples. By and large, these results should be useful in the field of complex systems (e.g., personalized medical treatment, social policy designing) to improve the interpretability/explainability in machine learning systems when deployed in real-world settings. Our work brings together two prominent fields in machine learning: ERM and causal inference, where the former is suitable to estimate high-dimensional functionals, while the latter is useful to determine which functional should be estimated that attains causal semantics.

## Footnotes

[1]That the r.h.s of Eq. (1) is independent of the value $r$ is known as a Verma constraint on the observed distribution implied by the causal graph [50].

[2]We refer to [38, 54] for the concept of pseudo-dimension.

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
