[Supplementary Material]

# Appendix – "Learning Causal Effects via Weighted Empirical Risk Minimization"

**Notations.** The following notations are used throughout this paper. Each variable will be represented with a capital letter $(X)$ and its realized value with the small letter $(x)$. We will use bold letters $(\mathbf{X})$ to denote sets of variables. Given an ordered set of variables $\mathbf{X} : X_1 < \cdots < X_n$, we denote $\mathbf{X}^{(i)} = \{X_1, \cdots, X_i\}$, and $\mathbf{X}^{\geq i} = \{X_i, \cdots, X_n\}$. We use the typical graph-theoretic terminology $PA(\mathbf{C})_G, Ch(\mathbf{C})_G, De(\mathbf{C})_G, An(\mathbf{C})_G$ to represent the union of $\mathbf{C}$ with its parents, children, descendants, ancestors in the graph $G$. We use $G_{\overline{\mathbf{C}_1}\underline{\mathbf{C}_2}}$ to denote the graph resulting from deleting all incoming edges to $\mathbf{C}_1$ and outgoing edges from $\overline{\mathbf{C}_2}$ in $G$. $G_{\mathbf{C}}$ denotes the subgraph of $G$ over $\mathbf{C}$. $(\mathbf{X} \perp\!\!\!\perp \mathbf{Y} \mid \mathbf{Z})_G$ denotes that $\mathbf{X}$ is d-separated from $\mathbf{Y}$ given $\mathbf{Z}$ in $G$. $\mathbb{E}_{P(\mathbf{y}|\mathbf{x})}[f(\mathbf{Y})|\mathbf{x}]$ denotes the conditional expectation of $f(\mathbf{Y})$ over $P(\mathbf{y}|\mathbf{x})$. $\mathcal{D} \equiv \{\mathbf{V}_{(i)}\}_{i=1}^m$ denotes a sample drawn from $P(\mathbf{v})$ where $\mathbf{V}_{(i)}$ denotes the $i$th sample in $\mathcal{D}$. The indicator function for $\mathbf{V}_{(i)} = \mathbf{v}$ is written as $I_{\mathbf{v}}(\mathbf{V}_{(i)})$. $P_m(\mathbf{v}) \equiv \frac{1}{m} \sum_{i=1}^m I_{\mathbf{v}}(\mathbf{V}_{(i)})$ denotes the empirical distribution of $\mathcal{D}$.

## A   Demonstrations of `wID` (Algorithm 1)

We demonstrate the application of Algo. 1 using Examples 1 (Fig. 1b), 2 (Fig. 2a), and 3 (Fig. 2b). First we restate `wID` algorithm and Lemma 1.

**Lemma A.1 (Restated Lemma 1).** *Let a topological order over* $\mathbf{V}$ *be* $V_1 < V_2 < \cdots < V_n$. *Suppose* $Q[\mathbf{A}]$ *is given by* $Q[\mathbf{A}] = P^{\mathcal{W}}(\mathbf{a}|\mathbf{r})$ *for some* $\mathbf{R} \subseteq \mathbf{V}$ *and weight function* $\mathcal{W}$.

*1. If* $\mathbf{W}$ *is a C-component of* $G_{\mathbf{A}}$, *then* $Q[\mathbf{W}] = P^{\mathcal{W}\times\mathcal{W}'}(\mathbf{w}|\mathbf{r}')$, *where* $\mathbf{R}' \equiv \mathbf{R} \cup ((\mathbf{A}\backslash\mathbf{W}) \cap An(\mathbf{W}))$ *and* $\mathcal{W}' \equiv \frac{P^{\mathcal{W}}((\mathbf{a}\backslash\mathbf{w})\cap An(\mathbf{w})|\mathbf{r})}{\prod_{V_i \in (\mathbf{A}\backslash\mathbf{W})\cap An(\mathbf{W})} P^{\mathcal{W}}(v_i|\mathbf{v}^{(i-1)}\cap \mathbf{a}\cap An(\mathbf{w}),\mathbf{r})}$.

*2. If* $\mathbf{W} \subseteq \mathbf{A}$ *satisfies* $\mathbf{W} = An(\mathbf{W})_{G_{\mathbf{A}}}$, *then* $Q[\mathbf{W}] = P^{\mathcal{W}}(\mathbf{w}|\mathbf{r})$.

**Example 1 (Figure 1b)**   Consider the model in Fig. 1b, where the causal effect is given by

$$P(y|do(x)) = \frac{\sum_w P(x,y|r,w)P(w)}{\sum_w P(x|r,w)P(w)}, \tag{A.1}$$

which is not in the weighting form. The graph has two C-components $\mathbf{S}_1 = \{W, X, Y\}$ and $\mathbf{S}_2 = \{R\}$ (Line 2). We have $Q[\mathbf{S}_1] = P^{\mathcal{W}_1}(\mathbf{s}_1|r)$ where $\mathcal{W}_1 = P(r)/P(r|w)$, and $Q[\mathbf{S}_2] = P(r|w)$ by Lemma 1 (Line 3). Let $\mathbf{D} = An(Y)_{G_{\mathbf{V}\backslash X}} = \{Y\}$ (Line 4). Run `wIdentify`$(Y, \mathbf{S}_1, Q[\mathbf{S}_1], r, \mathcal{W}_1)$ (Line 6). In Procedure `wIdentify`(), let $\mathbf{A} = An(Y)_{G_{\mathbf{S}_1}} = \{X, Y\}$, then $Q[\mathbf{A}] = P^{\mathcal{W}_1}(\mathbf{a}|r)$ (Line a.1). In $G_{\mathbf{A}} = G_{\{X,Y\}}$, let $\mathbf{S} \equiv \{Y\}$ denote the C-component containing $Y$ (Line a.5). Then, $Q[\mathbf{S}] = Q[Y] = P^{\mathcal{W}_1\times\mathcal{W}'}(y|r')$ where $\mathbf{R}' = \{R, X\}$ and $\mathcal{W}' = P^{\mathcal{W}}(x|r)/P^{\mathcal{W}}(x|r) = 1$ by Lemma 1 (with $\mathbf{W} = \mathbf{S} = Y$) (Line a.6). Line a.7 returns $Q[Y] = $ `wIdentify`$(Y, \mathbf{S}, Q[\mathbf{S}], r', \mathcal{W}_1) = P^{\mathcal{W}_1}(y|x, r)$. Finally we obtain $P(y|do(x)) = P^{\mathcal{W}_1}(y|x, r)$ (Line 7).

**Example 2 (Figure 2a)**   Consider Fig. 2a where the causal effect is given by

$$P(y|do(x)) = \sum_{w,z} P(z|w,x) \sum_{x'} P(y|w,x',z)P(x'|w)P(w). \tag{A.2}$$

**Algorithm A.1:** wID $(\mathbf{x}, \mathbf{y}, G, P)$ – Restated Algo. 1.

**Input:** $\mathbf{x}, \mathbf{y}, G, P$
**Output:** Expression of $P(\mathbf{y}|do(\mathbf{x}))$ as a weighted distribution; or FAIL if $P(\mathbf{y}|do(\mathbf{x}))$ is unidentifiable.

1 Let $\mathbf{V} \leftarrow An(\mathbf{Y})$; $P(\mathbf{v}) \leftarrow P(An(\mathbf{Y}))$; and $G \leftarrow G_{An(\mathbf{Y})}$.
2 Find the $C$-components of $G$: $\mathbf{S}_1, \cdots, \mathbf{S}_k$.
3 Let $Q[\mathbf{S}_i] = P^{\mathcal{W}_{\mathbf{s}_i}}(\mathbf{s}_i|\mathbf{r}_{\mathbf{s}_i})$ where $(\mathcal{W}_{\mathbf{s}_i}, \mathbf{r}_{\mathbf{s}_i})$ are derived from Lemma 1.
4 Let $\mathbf{D} \equiv An(\mathbf{Y})_{G_{\mathbf{V}\setminus\mathbf{X}}}$.
5 Find the $C$-component of $G_{\mathbf{D}}$: $\mathbf{D}_1, \cdots \mathbf{D}_K$.
6 For each $\mathbf{D}_i \in \mathbf{S}_j$ for some $(i,j)$, let
   $\quad Q[\mathbf{D}_i] = \mathtt{wIdentify}\left(\mathbf{D}_i, \mathbf{S}_j, Q[\mathbf{S}_j], \mathbf{r}_{\mathbf{s}_j}, \mathcal{W}_{\mathbf{s}_j}\right) \equiv P^{\mathcal{W}_{\mathbf{d}_i}}(\mathbf{d}_i|\mathbf{r}_{\mathbf{d}_i})$.
7 **if** $K = 1$ **then**
   $\quad$ **return** $P(\mathbf{y}|do(\mathbf{x})) = P^{\mathcal{W}_{\mathbf{d}_1}}(\mathbf{y}|\mathbf{r}_{\mathbf{d}_1})$.
   **end**
8 Let $\mathcal{W} \equiv \prod_{i=1}^{K} P^{\mathcal{W}_{\mathbf{d}_i}}(\mathbf{d}_i|\mathbf{r}_{\mathbf{d}_i})/P(\mathbf{d}|\mathbf{r})$ where $\mathbf{R} \equiv \mathbf{V}\setminus\mathbf{D}$.
9 **return** $P(\mathbf{y}|do(\mathbf{x})) = P^{\mathcal{W}}(\mathbf{y}|\mathbf{r})$

**Procedure** $\mathtt{wIdentify}(\mathbf{C}, \mathbf{T}, Q[\mathbf{T}], \mathbf{r}, \mathcal{W})$
$\quad$ **Input:** $\mathbf{T}, Q[\mathbf{T}] = P^{\mathcal{W}}(\mathbf{t}|\mathbf{r})$
$\quad$ **Output:** $Q[\mathbf{C}]$ for $\mathbf{C} \subseteq \mathbf{T}$ as a weighted distribution.
a.1 $\quad$ Let $\mathbf{A} \equiv An(\mathbf{C})_{G_{\mathbf{T}}}$, then $Q[\mathbf{A}] = P^{\mathcal{W}}(\mathbf{a}|\mathbf{r})$ by Lemma 1.
a.2 $\quad$ **if** $\mathbf{A} = \mathbf{C}$ **then**
   $\quad\quad$ **return** $Q[\mathbf{C}] = P^{\mathcal{W}}(\mathbf{a}|\mathbf{r})$
   $\quad$ **end**
a.3 $\quad$ **if** $\mathbf{A} = \mathbf{T}$ **then**
   $\quad\quad$ **return** *FAIL*
   $\quad$ **end**
a.4 $\quad$ **else**
a.5 $\quad\quad$ Let $\mathbf{S}$ denote the $C$-component in $G_{\mathbf{A}}$ such that $\mathbf{C} \subseteq \mathbf{S}$.
a.6 $\quad\quad$ Compute $Q[\mathbf{S}] = P^{\mathcal{W}\times\mathcal{W}'}(\mathbf{s}|\mathbf{r}')$ where $(\mathcal{W}', \mathbf{r}')$ are derived by Lemma 1.
a.7 $\quad\quad$ **return** $\mathtt{wIdentify}(\mathbf{C}, \mathbf{S}, Q[\mathbf{S}], \mathbf{r}', \mathcal{W}\times\mathcal{W}')$
   $\quad$ **end**

We start with $\mathbf{S}_1 = \{W, X, Y\}$ and $\mathbf{S}_2 = \{Z\}$ (Line 2). We then derive $Q[\mathbf{S}_1] = P^{\mathcal{W}_{\mathbf{S}_1}}(\mathbf{s}_1|z)$ where $\mathcal{W}_{\mathbf{S}_1} = P(z)/P(z|w,x)$ by applying Lemma 1 with $\mathbf{A} = \mathbf{V}$ and $\mathbf{W} = \mathbf{S}_1$ (Line 3). We also derive $Q[\mathbf{S}_2] = P^{\mathcal{W}_{\mathbf{S}_2}}(\mathbf{s}_2|x,w) = P(z|x,w)$ (where $\mathcal{W}_{\mathbf{S}_2} = 1$) by applying Lemma 1 with $\mathbf{A} = \mathbf{V}$ and $\mathbf{W} = \mathbf{S}_2$ (Line 3). Let $\mathbf{D} = An(Y)_{G_{\mathbf{V}\setminus X}} = \{W, Y, Z\}$ (Line 4), where $\mathbf{D}_1 = \{W, Y\}$ and $\mathbf{D}_2 = \{Z\}$ (Line 5).

For identifying $Q[\mathbf{D}_1]$, we invoke $\mathtt{wIdentify}(\mathbf{D}_1, \mathbf{S}_1, Q[\mathbf{S}_1], z, \mathcal{W}_{\mathbf{S}_1})$ (Line 6). Let $\mathbf{A}_1 = An(\mathbf{D}_1)_{G_{\mathbf{S}_1}} = \mathbf{D}_1$, then $Q[\mathbf{A}_1] = Q[\mathbf{D}_1] = P^{\mathcal{W}_{\mathbf{S}_1}}(\mathbf{d}_1|z)$ (Line a.1). Since $\mathbf{A}_1 = \mathbf{D}_1$, then we return $Q[\mathbf{D}_1] = P^{\mathcal{W}_{\mathbf{D}_1}}(\mathbf{d}_1|z)$ where $\mathcal{W}_{\mathbf{D}_1} = \mathcal{W}_{\mathbf{S}_1} = P(z)/P(z|w,x)$ (Line a.2).

For identifying $Q[\mathbf{D}_2]$, we invoke $\mathtt{wIdentify}(\mathbf{D}_2, \mathbf{S}_2, Q[\mathbf{S}_2], (w,x), 1)$ (Line 6). Let $\mathbf{A}_2 = An(\mathbf{D}_2)_{G_{\mathbf{S}_2}} = \mathbf{D}_2$, then $Q[\mathbf{D}_2] = P(\mathbf{d}_2|w,x)$ (Line a.1). Since $\mathbf{A}_2 = \mathbf{D}_2$, then we return $Q[\mathbf{D}_2] = P^{\mathcal{W}_{\mathbf{D}_2}}(\mathbf{d}_2|x,w) = P(z|x,w)$ where $\mathcal{W}_{\mathbf{D}_2} = 1$ (Line a.2).

Let $\mathcal{W} \equiv P^{\mathcal{W}_{\mathbf{D}_1}}(\mathbf{d}_1|z) P^{\mathcal{W}_{\mathbf{D}_2}}(\mathbf{d}_2|x,w)/P(\mathbf{d}|x)$ (Line 8). Specifically,

$$\mathcal{W} \equiv P^{\mathcal{W}_{\mathbf{D}_1}}(\mathbf{d}_1|z) P^{\mathcal{W}_{\mathbf{D}_2}}(\mathbf{d}_2|x,w)/P(\mathbf{d}|x)$$

$$= \frac{P^{\mathcal{W}_{\mathbf{D}_1}}(w,y|z) P(z|x,w)}{P(w,z,y|x)}$$

Finally, the causal effect is given by $P(y|do(x)) = P^{\mathcal{W}}(y|x)$ (Line 9).

**Example 3 (Figure 2b)** Consider Fig. 2b where the causal effect is given by

$$P(y|do(x,r)) = \sum_{w,z} P(z|w,x) \sum_{x'} P(y|w,x',r,z)P(x'|w,r)P(w). \quad\quad (A.3)$$

We start with $\mathbf{S}_1 = \{W, X, Y\}$ and $\mathbf{S}_2 = \{R, Z\}$ (Line 2). We then derive $Q[\mathbf{S}_1] = P^{\mathcal{W}_{\mathbf{S}_1}}(\mathbf{s}_1|r,z)$ where $\mathcal{W}_{\mathbf{S}_1} = P(r,z)/P(z|w,x,r)P(r|w)$ by applying Lemma 1 with $\mathbf{A} = \mathbf{V}$ and $\mathbf{W} = \mathbf{S}_1$ (Line 3). We also derive $Q[\mathbf{S}_2] = P^{\mathcal{W}_{\mathbf{S}_2}}(\mathbf{s}_2|x,w) = P(z|x,w)$ (where $\mathcal{W}_{\mathbf{S}_2} = 1$) by applying Lemma 1 with $\mathbf{A} = \mathbf{V}$ and $\mathbf{W} = \mathbf{S}_2$ (Line 3). Let $\mathbf{D} = An(Y)_{G_{\mathbf{V}\setminus\{X,R\}}} = \{W, Y, Z\}$ (Line 4), where $\mathbf{D}_1 = \{W, Y\}$ and $\mathbf{D}_2 = \{Z\}$ (Line 5).

For identifying $Q\left[\mathbf{D}_1\right]$, we invoke $\mathtt{wIdentify}\left(\mathbf{D}_1, \mathbf{S}_1, Q\left[\mathbf{S}_1\right], \{r, z\}, \mathcal{W}_{\mathbf{S}_1}\right)$ (Line 6). Let $\mathbf{A}_1 = An(\mathbf{D}_1)_{G_{\mathbf{S}_1}} = \mathbf{D}_1$, then $Q\left[\mathbf{A}_1\right] = P^{\mathcal{W}_{\mathbf{S}_1}}\left(\mathbf{a}_1|r, z\right)$ by applying Lemma 1 (Line a.1). Since $\mathbf{A}_1 = \mathbf{D}_1$, then we return $Q\left[\mathbf{D}_1\right] = P^{\mathcal{W}_{\mathbf{D}_1}}\left(\mathbf{d}_1|r, z\right)$ where $\mathcal{W}_{\mathbf{D}_1} = \mathcal{W}_{\mathbf{S}_1}$ (Line a.2).

For identifying $Q\left[\mathbf{D}_2\right]$, we invoke $\mathtt{wIdentify}\left(\mathbf{D}_2, \mathbf{S}_2, Q\left[\mathbf{S}_2\right], (w, x), 1\right)$ (Line 6). Let $\mathbf{A}_2 = An(\mathbf{D}_2)_{G_{\mathbf{S}_2}} = \mathbf{D}_2$, then $Q\left[\mathbf{A}_2\right] = P^{\mathcal{W}_{\mathbf{S}_2}}\left(\mathbf{a}_2|w, x\right) = P(\mathbf{d}_2|w, x)$ by Lemma 1 (Line a.1). Since $\mathbf{A}_2 = \mathbf{D}_2$, then we return $Q\left[\mathbf{D}_2\right] = P^{\mathcal{W}_{\mathbf{D}_2}}\left(\mathbf{d}_2|x, w\right) = P(z|x, w)$ where $\mathcal{W}_{\mathbf{D}_2} = 1$ (Line a.2).

Let $\mathcal{W} \equiv P^{\mathcal{W}_{\mathbf{D}_1}}\left(\mathbf{d}_1|r, z\right) P^{\mathcal{W}_{\mathbf{D}_2}}\left(\mathbf{d}_2|x, w\right)/P(\mathbf{d}|x, r)$ (Line 8). Specifically,

$$\mathcal{W} \equiv P^{\mathcal{W}_{\mathbf{D}_1}}\left(\mathbf{d}_1|r, z\right) P^{\mathcal{W}_{\mathbf{D}_2}}\left(\mathbf{d}_2|x, w\right)/P(\mathbf{d}|r, x)$$
$$= \frac{P^{\mathcal{W}_{\mathbf{D}_1}}\left(w, y|r, z\right) P(z|x, w)}{P(w, z, y|r, x)}$$

Finally, the causal effect is given by $P\left(y|do(x, r)\right) = P^{\mathcal{W}}\left(y|x, r\right)$ (Line 9).

**Remark: The use of extra covariates in Algo. 1.** We note that the result of Algo. 1 is given by $P\left(\mathbf{y}|do(\mathbf{x})\right) = P^{\mathcal{W}}\left(\mathbf{y}|\mathbf{r}\right)$ for some $\mathbf{R} \supseteq \mathbf{X}$, despite that $P\left(\mathbf{y}|do(\mathbf{x})\right)$ should be a function of only $\mathbf{X} = \mathbf{x}$ instead of $\mathbf{R} = \mathbf{r}$. For instance, in Example 1 (Figure 1b), we obtain $P\left(y|do(x)\right) = P^{\mathcal{W}}\left(y|x, r\right)$. That $P^{\mathcal{W}}\left(y|x, r\right)$ is independent of the value $r$, or equivalently, the r.h.s of Eq. (A.1) is independent of the value $r$, is known as a *Verma constraint* on the observed distribution implied by the causal graph [7]. Despite the equality $P^{\mathcal{W}}\left(\mathbf{y}|\mathbf{r}\right) = P^{\mathcal{W}}\left(\mathbf{y}|\mathbf{x}\right)$ by Verma constraints, we use the estimand $P^{\mathcal{W}}\left(\mathbf{y}|\mathbf{r}\right)$ instead of $P^{\mathcal{W}}\left(\mathbf{y}|\mathbf{x}\right)$ in finite sample settings, since the inclusion of more covariates tends to reduce the error in the regression analysis [1].

# B   Procedure for Evaluating Weight Function $\widehat{\mathcal{W}^*}$ in $\mathtt{WERM\text{-}ID\text{-}R}$ (Algorithm 2)

Notice that Algo. 1 computes $\mathcal{W}^*$ (i.e. $\mathcal{W}$ in Line 8) and expresses a causal estimand into a weighted distribution recursively by repeated application of Lemma 1. Given finite samples $\mathcal{D} = \{\mathbf{V}_{(i)}\}_{i=1}^m$ drawn from $P(\mathbf{v})$, one can evaluate $\widehat{\mathcal{W}^*}$ by running $\mathtt{wID}$ (Algo. 1) and computing weights recursively if we can evaluate the weights in Lemma 1 from $\mathcal{D} = \{\mathbf{V}_{(i)}\}_{i=1}^m$. We provide a procedure $\mathtt{LearnWeightedDist}$ given in Algo. B.1 for evaluating $\mathcal{W} \times \mathcal{W}'$ in Lemma 1 when given $\mathcal{D} \sim P(\mathbf{v})$ and the weights $\mathcal{W}$. The key idea is that $\widehat{P}^{\mathcal{W}}(\cdot|\cdot)$ will be computed by drawing samples $\mathcal{D}^{\mathcal{W}}$ that could be treated as if they were drawn from $P^{\mathcal{W}}(\mathbf{v})$ in asymptotic. Specifically, $\mathtt{LearnWeightedDist}$ evaluates $\mathcal{W}'$ in Lemma 1 from $\mathcal{D}^{\mathcal{W}}$, generates samples $\mathcal{D}^{\mathcal{W} \times \mathcal{W}'}$ by weighting $\mathcal{D}$ with $\mathcal{W} \times \mathcal{W}'$ using a procedure $\mathtt{WeightedSample}$, and outputs $(\mathcal{W} \times \mathcal{W}', \mathcal{D}^{\mathcal{W} \times \mathcal{W}'})$. The procedure $\mathtt{WeightedSample}(\mathcal{D}, \mathcal{W})$ draws sample $\mathcal{D}^{\mathcal{W}}$ based on $\mathcal{D}$ by repeatedly taking bootstrap samples $\mathcal{D}'$ from $\mathcal{D}$ and re-sampling $\mathcal{D}'$ with the weight $\mathcal{W}$.

Given a weight function $\mathcal{W}$, let $P_m^{\mathcal{W}}(\mathbf{v})$ denote the normalized empirical distribution $P_m(\mathbf{v})$ of $\mathcal{D} = \{\mathbf{V}_{(i)}\}_{i=1}^m$ weighted by $\mathcal{W}$, i.e.,

$$P_m^{\mathcal{W}}(\mathbf{v}) \equiv \frac{\mathcal{W}(\mathbf{v})P_m(\mathbf{v})}{\sum_{\mathbf{v}} \mathcal{W}(\mathbf{v})P_m(\mathbf{v})}. \tag{B.4}$$

The following results ascertain that (1) $\mathcal{D}^{\mathcal{W}}$ output by $\mathtt{WeightedSample}(\mathcal{D}, \mathcal{W})$ are samples that could be treated as those drawn from $P^{\mathcal{W}}(\mathbf{v})$ in asymptotic; and (2) The probability of $|\mathcal{D}^{\mathcal{W}}| \geq \mathcal{D}$ is extremely high. For example, if $a = 5, m = 100$, then the probability $|\mathcal{D}^{\mathcal{W}}| < |\mathcal{D}|$ is smaller than $10^{-70}$.

**Lemma B.1 (Correctness of $\mathtt{WeightedSample}$ in Algo. B.1).** *Let $\mathbf{V}_{(j)} \in \mathcal{D}^{\mathcal{W}}$ denote the jth sample of $\mathcal{D}^{\mathcal{W}}$, the set of samples returned by $\mathtt{WeightedSample}(\mathcal{D}, \mathcal{W})$ in Algo. B.1. Then, (1) $\mathcal{D}^{\mathcal{W}}$ follows the distribution $P_m^{\mathcal{W}}(\mathbf{v})$; (2) $P_m^{\mathcal{W}}(\mathbf{v})$ converges to $P^{\mathcal{W}}(\mathbf{v})$ for all $\mathbf{v}$ as $m \to \infty$; and (3) $P(|\mathcal{D}^{\mathcal{W}}| \geq |\mathcal{D}|) \geq 1 - \exp\left(-0.5(1 - 1/a)^2 am\right)$.*

*Proof.* In the proof, we will use $Pr(\cdot)$ to denote any probability measure assigned to any event in the subset of sample spaces.

---

**Algorithm B.1:** `LearnWeightedDist`$(\mathcal{D}, \mathcal{D}^{\mathcal{W}}, \mathcal{W}, (\mathbf{A}, \mathbf{W}, \mathbf{R}))$ –Evaluating weights in Lemma 1

---

**Input:** Samples $\mathcal{D} = \{\mathbf{V}_{(i)}\}_{i=1}^{m}$ drawn from $P(\mathbf{v})$; Estimated weight $\mathcal{W}$; Samples $\mathcal{D}^{\mathcal{W}}$ drawn from $P_m^{\mathcal{W}}(\mathbf{v})$.

**Output:** Estimated weights $\mathcal{W} \times \widehat{\mathcal{W}'}$; Samples $\mathcal{D}^{\mathcal{W} \times \widehat{\mathcal{W}'}}$ drawn from $P_m^{\mathcal{W} \times \widehat{\mathcal{W}'}}(\mathbf{v})$.

1  Evaluate $\widehat{\mathcal{W}'} \equiv \dfrac{\widehat{P}^{\mathcal{W}}((\mathbf{a} \backslash \mathbf{w}) \cap An(\mathbf{w}) | \mathbf{r})}{\prod_{V_k \in (\mathbf{A} \backslash \mathbf{W}) \cap An(\mathbf{W})} \widehat{P}^{\mathcal{W}}(v_k | \mathbf{v}^{(k-1)} \cap \mathbf{a} \cap An(\mathbf{w}), \mathbf{r})}$ by computing $\widehat{P}^{\mathcal{W}}(\cdot | \cdot)$ from samples $\mathcal{D}^{\mathcal{W}}$
   using regressions.

2  Evaluate $\mathcal{W} \times \widehat{\mathcal{W}'}$.

3  Generate $\mathcal{D}^{\mathcal{W} \times \widehat{\mathcal{W}'}} = $ `WeightedSample`$(\mathcal{D}, \mathcal{W} \times \widehat{\mathcal{W}'})$.

4  **return** $(\mathcal{W} \times \widehat{\mathcal{W}'}, \mathcal{D}^{\mathcal{W} \times \widehat{\mathcal{W}'}})$

**Procedure** `WeightedSample`$(\mathcal{D}, \mathcal{W})$
  **Input:** Samples $\mathcal{D}$ drawn from $P(\mathbf{v})$; A weight function $\mathcal{W}(\mathbf{v})$.
  **Output:** Samples $\mathcal{D}^{\mathcal{W}}$ drawn from $P_m^{\mathcal{W}}(\mathbf{v})$.

1  $\quad \mathcal{D}^{\mathcal{W}} = \{\}$.

2  $\quad$ Let $\mathcal{W}_{\max} \equiv \max\left\{1, \max_{\mathbf{V}_{(j)} \in \mathcal{D}} \mathcal{W}(\mathbf{V}_{(j)})\right\}$.

3  $\quad$ Let $j = 0$ and $j_{\max} \equiv a \lceil \mathcal{W}_{\max} \rceil$ for some constant $a \geq 2$. // `e.g.,` $a = 10$
  $\quad$ **while** $|\mathcal{D}^{\mathcal{W}}| < \mathcal{D}$ **do**
4  $\quad\quad$ $j = j + 1$.
5  $\quad\quad$ Take a bootstrap sampling $\mathcal{D}'$ from $\mathcal{D}$.
6  $\quad\quad$ **for** $i = 1, 2, \cdots, |\mathcal{D}'|$ **do**
7  $\quad\quad\quad$ Generate $A_{(i)}$ from $P(A_{(i)} = 1 | \mathbf{V}_{(i)}) \equiv$ Bernoulli $\left(\frac{\mathcal{W}(\mathbf{V}_{(i)})}{\mathcal{W}_{\max}}\right)$ where $\mathbf{V}_{(i)} \in \mathcal{D}'$.
   $\quad\quad\quad$ // `Bernoulli`$(\theta)$ `is a Bernoulli distribution parameterized by`
   $\quad\quad\quad$ $\theta \in [0, 1]$.
8  $\quad\quad\quad$ If $A_{(i)} = 1$, then $\mathcal{D}^{\mathcal{W}} = \mathcal{D}^{\mathcal{W}} \cup \{\mathbf{V}_{(i)}\}$.
   $\quad\quad$ **end**
9  $\quad\quad$ **if** $j > j_{\max}$ **then end loop**
  $\quad$ **end**
10 **return** $\mathcal{D}^{\mathcal{W}}$

---

We note that the samples of $\mathcal{D}^{\mathcal{W}}$ are chosen from $\mathcal{D}'$, which was collected through the bootstrapped sampling from $\mathcal{D}$. Note that the bootstrapped samples $\mathcal{D}'$ follow the empirical distribution of $\mathcal{D}$, denoted as $P_m$, i.e., $\mathcal{D}' \sim P_m$. Let $\mathcal{D}^{\mathcal{W}} = \{\mathbf{V}_{(i)}^{\mathcal{W}}\}_{i=1}^{m'}$ and $\mathcal{D} = \{\mathbf{V}_{(i)}\}_{i=1}^{m}$. By the design of Algo. B.1, we note $Pr(A_{(i)} = 1|\mathbf{v}) = \frac{\mathcal{W}(\mathbf{v})}{\mathcal{W}_{\max}}$; $Pr(\mathbf{V}_{(i)} = \mathbf{v}) = P_m(\mathbf{v})$ (where $\mathbf{V}_{(i)} \in \mathcal{D}$). Then, for $\mathbf{V}_{(i)}^{\mathcal{W}} \in \mathcal{D}^{\mathcal{W}}$,

$$
\begin{aligned}
Pr(\mathbf{V}_{(i)}^{\mathcal{W}} = \mathbf{v}) &= Pr(\mathbf{V}_{(i)} = \mathbf{v}|A_{(i)} = 1) \\
&= \frac{Pr(A_{(i)} = 1|\mathbf{V}_{(i)} = \mathbf{v})Pr(\mathbf{V}_{(i)} = \mathbf{v})}{\sum_{\mathbf{v}} Pr(A_{(i)} = 1|\mathbf{V}_{(i)} = \mathbf{v})Pr(\mathbf{V}_{(i)} = \mathbf{v})} \\
&= \frac{Pr(A_{(i)} = 1|\mathbf{v})P_m(\mathbf{v})}{\sum_{\mathbf{v}} Pr(A_{(i)} = 1|\mathbf{v})P_m(\mathbf{v})} \\
&= \frac{P_m(\mathbf{v})\mathcal{W}(\mathbf{v})/\mathcal{W}_{\max}}{\sum_{\mathbf{v}} P_m(\mathbf{v})\mathcal{W}(\mathbf{v})/\mathcal{W}_{\max}} \\
&= \frac{P_m(\mathbf{v})\mathcal{W}(\mathbf{v})}{\sum_{\mathbf{v}} P_m(\mathbf{v})\mathcal{W}(\mathbf{v})} \\
&= P_m^{\mathcal{W}}(\mathbf{v}),
\end{aligned}
$$

To see the second statement holds, we note that $\lim_{m \to \infty} P_m(\mathbf{v}) = P(\mathbf{v})$ for any possible realization of $\mathbf{V} = \mathbf{v}$ by the Strong law of large number. Then, $\lim_{m \to \infty} P_m(\mathbf{v})\mathcal{W}(\mathbf{v}) = P(\mathbf{v})\mathcal{W}(\mathbf{v})$. Now,

consider the following:

$$\lim_{m \to \infty} \frac{\mathcal{W}P_m(\mathbf{v})}{\sum_{\mathbf{v}} \mathcal{W}(\mathbf{v})P_m(\mathbf{v})} = \frac{\mathcal{W}(\mathbf{v})P(\mathbf{v})}{\sum_{\mathbf{v}} \mathcal{W}(\mathbf{v})P(\mathbf{v})} \tag{B.5}$$

$$= \mathcal{W}(\mathbf{v})P(\mathbf{v}) \tag{B.6}$$

$$= P^{\mathcal{W}}(\mathbf{v}), \tag{B.7}$$

where the first equality holds since $\frac{\mathcal{W}P_m(\mathbf{v})}{\sum_{\mathbf{v}} \mathcal{W}(\mathbf{v})P_m(\mathbf{v})}$ is continuous with respect to $P_m$ whenver $\mathcal{W} > 0$ and $\mathcal{W} < \infty$; the second equality holds since $\sum_{\mathbf{v}} \mathcal{W}(\mathbf{v})P(\mathbf{v}) = \sum_{\mathbf{v}} P^{\mathcal{W}}(\mathbf{v}) = 1$ by the definition of the weight (Def. 1); and third equality holds by the definition of the weighted distribution.

To see the third statement holds, proving that the stopping condition $j > j_{\max}$ happens at exteremely low probability is sufficient. Let the number of samples of $\mathcal{D}^{\mathcal{W}}$ collected at $j$th iteration be $M_j \equiv \sum_{i=1}^{|\mathcal{D}'|} A_{(i)}$. We note $\mu_M \equiv \mathbb{E}[M_j] = m/\mathcal{W}_{\max}$ (for all $j$) since $P(A_{(i)} = 1) = 1/\mathcal{W}_{\max}$ for all $i = 1, 2, \cdots, |\mathcal{D}'|$. When the algorithm terminates, the number of collected samples are $S \equiv M_1 + M_2 + \cdots + M_{j_{\max}}$ and $\mu_S \equiv \mathbb{E}[S] = j_{\max}\mu_M = a\lceil \mathcal{W}_{\max}\rceil \mu_M \geq am$. By applying Chernoff bound, $P(S < (1-\delta)\mu_S) \leq \exp(-0.5\delta^2 \mu_S) \leq \exp(-0.5\delta^2 am)$ for $\delta \in [0, 1]$. By fixing $(1-\delta) = \frac{\mathcal{W}_{\max}}{a\lceil \mathcal{W}_{\max}\rceil}$, we derive $P(S < m) \leq \exp(-0.5\delta^2 am)$. Since $\delta \geq (1 - 1/a)$, we conclude $P(S < m) \leq \exp(-0.5(1 - 1/a)^2 am)$. This completes the proof. $\square$

The asymptotic correctness of the procedure `LearnWeightedDist` is guaranteed by the following:

**Lemma B.2 (Correctness of `LearnWeightedDist` (Algo. B.1)).** *Suppose $\widehat{P}^{\mathcal{W}}(\cdot|\cdot)$ in the computation of $\widehat{\mathcal{W}'}$ in Line 1 of `LearnWeightedDist` (Algo. B.1) is a correct estimate of $P_m^{\mathcal{W}}(\cdot|\cdot)$. Then, for $(\mathcal{W} \times \widehat{\mathcal{W}'}, \mathcal{D}^{\mathcal{W} \times \widehat{\mathcal{W}'}}) = \mathtt{LearnWeightedDist}(\mathcal{D}, \mathcal{D}^{\mathcal{W}}, \mathcal{W}, (\mathbf{A}, \mathbf{W}, \mathbf{R}))$, $\mathcal{W} \times \widehat{\mathcal{W}'}$ converges to $(\mathcal{W} \times \mathcal{W}')$ as $m \to \infty$ and $\mathcal{D}^{\mathcal{W} \times \widehat{\mathcal{W}'}}$ follows the true distribution $P^{\mathcal{W} \times \mathcal{W}'}(\mathbf{v})$ in the limit of infinite samples.*

*Proof.* From the given assumption, $\widehat{P}^{\mathcal{W}}(\cdot|\cdot)$ learned from $\mathcal{D}^{\mathcal{W}}$ are correct estimates of $P_m^{\mathcal{W}}(\cdot|\cdot)$. This implies $\widehat{\mathcal{W}'} = \frac{P_m^{\mathcal{W}}((\mathbf{a}\backslash\mathbf{w}) \cap An(\mathbf{w})|\mathbf{r})}{\prod_{V_k \in (\mathbf{A}\backslash\mathbf{w}) \cap An(\mathbf{w})} P_m^{\mathcal{W}}(v_k|\mathbf{v}^{(k-1)} \cap \mathbf{a} \cap An(\mathbf{w}), \mathbf{r})}$. By the second statement of Lemma B.1, which states $P_m^{\mathcal{W}}(\mathbf{v})$ converges to $P^{\mathcal{W}}(\mathbf{v})$ for all $\mathbf{v}$, $\mathcal{W} \times \widehat{\mathcal{W}'}$ converges to $(\mathcal{W} \times \mathcal{W}')$ as $m \to \infty$. Also, since $\mathcal{D}^{\mathcal{W} \times \widehat{\mathcal{W}'}}$ are samples drawn from $P_m^{\mathcal{W} \times \widehat{\mathcal{W}'}}$, in the limit of infinite samples, $\mathcal{D}^{\mathcal{W} \times \widehat{\mathcal{W}'}}$ follows the true distribution $P^{\mathcal{W} \times \mathcal{W}'}(\mathbf{v})$.

$\square$

The time complexity of `LearnWeightedDist` is given as follows:

**Lemma B.3 (Time complexity of Algo. B.1).** *Suppose $0 < \mathcal{W} \times \widehat{\mathcal{W}'} < c$ for some constant $c > 0$. Let $T_1(m)$ denote the time complexity for learning $\widehat{P}^{\mathcal{W}}(\cdot|\cdot)$ from samples $\mathcal{D}^{\mathcal{W}}$. Let $n \equiv |\mathbf{V}|$. Then, `LearnWeightedDist` (Algo. B.1) runs in $O(mc + nT_1(m))$ time.*

*Proof.* We first note that `WeightedSample`$(\mathcal{D}, \mathcal{W})$ takes $O(amc) = O(mc)$ since $\lceil \mathcal{W}_{\max}\rceil \leq c$. Line 1 of `LearnWeightedDist` takes $O(nT_1(m))$. Line 2 takes $O(m)$, since $|\mathcal{D}^{\mathcal{W}}| = O(m)$ by the `While` loop condition in `WeightedSample`. Line 3 takes $O(mc)$. Summing up, Algo. B.1 takes $O(nT_1(m) + m + mc) = O(nT_1(m) + mc)$.

$\square$

Equipped with `LearnWeightedDist` (Algo. B.1), we evaluate $\widehat{\mathcal{W}^*}$ by running `wID` (Algo. 1) while invoking `LearnWeightedDist` whenever `wID` calls Lemma 1. The time complexity of evaluating $\widehat{\mathcal{W}^*}$ is given as follows:

**Lemma B.4 (Time complexity for evaluating $\widehat{\mathcal{W}^*}$).** *Let $\mathcal{W}^*$ denote the weight estimand defined in Line 8 (or Line 7) of `wID` (Algo. 1) such that $P(\mathbf{y}|do(\mathbf{x})) = P^{\mathcal{W}^*}(\mathbf{y}|\mathbf{r})$. Let $n \equiv |\mathbf{V}|$. Let $K$ denote the number of $C$-components in $G_{\mathbf{D}}$ (in Algo. 1). Let $T_1(m)$ denote the time complexity for*

*learning $\widehat{P}^{\mathcal{W}}(\cdot|\cdot)$ from samples $\mathcal{D}^{\mathcal{W}}$. Assume all weights satisfy $0 < \mathcal{W} < c$ for some constant $c > 0$. Suppose we evaluate $\widehat{\mathcal{W}^*}$ by running `wID` and invoking `LearnWeightedDist` (Algo. B.1) whenever `wID` calls Lemma 1. Then, evaluating $\widehat{\mathcal{W}^*}$ takes $O\left(nK\left(mc + nT_1(m)\right)\right)$.*

*Proof.* We note that the number of $C$-components of $G_{\mathbf{D}}$ is $K$. In identifying $Q\left[\mathbf{D}_i\right]$, `LearnWeightedDist` is called at most $n$ times. Therefore, by Lemma B.3, it takes $O(K \times n(mc + nT_1(m)))$ to evaluate $\widehat{\mathcal{W}^*}$ . $\hfill\square$

## C  Proofs

### C.1  Background Results

#### C.1.1  Multi-outcome Sequential Back-door (mSBD) Criterion

**Definition C.1 (Multi-outcome sequential back-door (mSBD) criterion** [3]**).** Given the pair of sets $(\mathbf{X}, \mathbf{Y})$, let $\mathbf{X} = \{X_1, X_2, \cdots, X_n\}$ be topologically ordered as $X_1 < X_2 < \cdots < X_n$. Let $\mathbf{Y}_0 = \mathbf{Y} \setminus De(\mathbf{X})$ and $\mathbf{Y}_i = \mathbf{Y} \cap \left(De(X_i) \setminus De(\mathbf{X}^{\geq i+1})\right)$ for $i = 1, \cdots, n$. A sequence $\mathbf{Z} = (\mathbf{Z}_1, \cdots, \mathbf{Z}_n)$ is mSBD admissible relative to $(\mathbf{X}, \mathbf{Y})$ if it holds that $\mathbf{Z}_i \subseteq ND\left(\mathbf{X}^{\geq i}\right)$, and $\left(\mathbf{Y}^{\geq i} \perp\!\!\!\perp X_i | \mathbf{Y}^{(i-1)}, \mathbf{Z}^{(i)}, \mathbf{X}^{(i-1)}\right)_{G_{\underline{X_i}\overline{\mathbf{X}^{\geq i+1}}}}$ for $i = 1, \cdots, n$.

**Theorem C.1 (mSBD adjustment [3, Thm. 1]).** *If $\mathbf{Z}$ is mSBD admissible relative to $(\mathbf{X}, \mathbf{Y})$, then $P\left(\mathbf{y}|do(\mathbf{x})\right)$ is identifiable and given by*

$$P\left(\mathbf{y}|do(\mathbf{x})\right) = \sum_{\mathbf{z}} \prod_{k=0}^{n} P\left(\mathbf{y}_k | \mathbf{x}^{(k)}, \mathbf{z}^{(k)}, \mathbf{y}^{(k-1)}\right) \times \prod_{j=1}^{n} P\left(\mathbf{z}_j | \mathbf{x}^{(j-1)}, \mathbf{z}^{(j-1)}, \mathbf{y}^{(j-1)}\right). \quad (\text{C.8})$$

**Theorem C.2 (Representation of mSBD adjustment as a weighted distribution [3, Thm. 2]).** *If $\mathbf{Z}$ is mSBD admissible relative to $(\mathbf{X}, \mathbf{Y})$, then*

$$P\left(\mathbf{y}|do(\mathbf{x})\right) = P^{\mathcal{W}}(\mathbf{y}|\mathbf{x}), \text{ where } \mathcal{W} = \frac{P(\mathbf{x})}{\prod_{k=1}^{n} P\left(x_k | \mathbf{x}^{(k-1)}, \mathbf{y}^{(k-1)}, \mathbf{z}^{(k)}\right)}. \quad (\text{C.9})$$

**Lemma C.1 (mSBD adjustment and $C$-factor identification).** *Let $\mathbf{S}$ denote a union of some $C$-components of $G$. If $\mathbf{W} \subseteq \mathbf{S}$ satisfies $\mathbf{W} = An(\mathbf{W})$ in $G_{\mathbf{S}}$, then (1) $(\mathbf{S}\setminus\mathbf{W}) \cap An(\mathbf{W})$ is mSBD admissible relative to $((\mathbf{V}\setminus\mathbf{S}) \cap An(\mathbf{W}), \mathbf{W})$; and (2) $P\left(\mathbf{w}|do(\mathbf{v}\setminus\mathbf{w})\right) = P\left(\mathbf{w}|do((\mathbf{v}\setminus\mathbf{s}) \cap An(\mathbf{w}))\right)$, which is identifiable by the mSBD adjustment by Thm. C.1.*

*Proof.* Two things that we will prove are following:

1. $(\mathbf{S}\setminus\mathbf{W}) \cap An(\mathbf{W})$ satisfies the mSBD criterion relative to $((\mathbf{V}\setminus\mathbf{S}) \cap An(\mathbf{W}), \mathbf{W})$; and

2. $P\left(\mathbf{w}|do((\mathbf{v}\setminus\mathbf{s}) \cap An(\mathbf{w}))\right) = P\left(\mathbf{w}|do(\mathbf{v}\setminus\mathbf{w})\right) = Q\left[\mathbf{W}\right]$.

We start by proving the **first statement**. For the notational convenience, let $\mathbf{Z} \equiv (\mathbf{S}\setminus\mathbf{W}) \cap An(\mathbf{W})$. Let $\mathbf{R} \equiv (\mathbf{V}\setminus\mathbf{S}) \cap An(\mathbf{W})$. Let $\mathbf{R} = \{R_1, R_2, \cdots, R_n\}$ where $R_1 \prec R_2 \prec \cdots \prec R_n$. Let $\mathbf{W}_0 = \mathbf{W}\setminus De(\mathbf{R})$, and $\mathbf{W}_i = \mathbf{W} \cap (De(R_i)\setminus De(\mathbf{R}^{\geq i+1}))$ for $i = 1, 2, \cdots, n$.

We first partition $\mathbf{Z} = \{\mathbf{Z}_1, \cdots, \mathbf{Z}_n\}$ as follow: $\mathbf{Z}_1 = \mathbf{Z} \cap ND(\mathbf{R})$, and $\mathbf{Z}_k \equiv (\mathbf{Z}\setminus\mathbf{Z}^{(k-1)}) \cap ND(\mathbf{R}^{\geq k})$. To witness that such partition is possible, it suffices to show that there exists no $Z_k \in \mathbf{Z}$ that is a descendent of $R_n$. Suppose there exists such $Z_k$; i.e., there exists a path $R_n \to \cdots \to Z_k$. Since $Z_k$ is an ancestor of some $W_j \in \mathbf{W}$, $Z_k \to \cdots \to W_j$. Note $W_j \in \mathbf{W}_n$ since $R_n \to \cdots \to Z_k \to \cdots \to W_j$. We note that there should be some variables $C_i \in \mathbf{V}\setminus\mathbf{S}$ on the path from $Z_k$ to $W_j$; Otherwise, all internal nodes on the path (other than $R_n$) belongs to $\mathbf{S}$, implying that $Z_k$ should be included in $\mathbf{W}$ (since $Z_k$ should be included in the ancestral set of $\mathbf{S}$), which is a contradiction. Suppose the path includes such $C_i$. This implies that $C_i$ is a parent of some nodes on $\mathbf{S}$, which contradicts that the path stems from $R_n$ such that $R_1 \prec \cdots \prec R_n$. Therefore, there are no such $Z_k$. This implies that we can partition $\mathbf{Z}$ as $\mathbf{Z} = \{\mathbf{Z}_1, \cdots, \mathbf{Z}_n\}$.

By such partition, the condition $\mathbf{Z}_i \subseteq ND(\mathbf{R}^{\geq i})$ is automatically satisfied. Thus, We focus on showing

$$\left(\mathbf{W}^{\geq i} \perp\!\!\!\perp R_i | \mathbf{W}^{(i-1)}, \mathbf{Z}^{(i)}, \mathbf{R}^{(i-1)}\right)_{G_{R_i \overline{\mathbf{R}^{\geq i+1}}}}. \tag{C.10}$$

On $G' \equiv G_{\underline{R_i}\overline{\mathbf{R}^{\geq i+1}}}$, we consider the latent projected graph $G'' \equiv G'[\mathbf{W}, \mathbf{R}^{(i)}, \mathbf{Z}^{(i)}]$ (i.e., the latent projection of $\mathbf{V}$ onto $\mathbf{W}, \mathbf{R}^{(i)}, \mathbf{Z}^{(i)}$ [4, Def. 1]) without loss of generality, since the projected graph preserves the independence between $\mathbf{W}, \mathbf{R}^{(i)}, \mathbf{Z}^{(i)}$ on $G'$. On $G''$, suppose there exists a path $p$ connecting $R_i \in \mathbf{R} = (\mathbf{V}\backslash\mathbf{S}) \cap An(\mathbf{W})$ to $W_j \in \mathbf{W}^{\geq i}$ conditioned on $\mathbf{W}^{(i-1)}, \mathbf{Z}^{(i)}, \mathbf{R}^{(i-1)}$.

The path has the following form. Let $R_j \in Pa(R_i)\backslash\{R_i\}$. Let $R_p \in An(R_i)\backslash Pa(R_i)$.

$$R_i\{\leftarrow \vee\{\leftrightarrow, \emptyset\}\}R_j\{\leftarrow \vee\{\leftrightarrow, \emptyset\}\}R_p\{\leftarrow \vee \rightarrow \vee\emptyset\}S_k\{\leftrightarrow \wedge\{\rightarrow \vee \leftarrow \vee\emptyset\}\}W_j,$$

where $S_k \in \mathbf{S}\backslash\{W_j\} \subseteq \mathbf{W}\cup\mathbf{Z}^{(i)}$. Suppose $R_p \leftarrow S_k$. This means that $S_k \in \left(\mathbf{W}\cup\mathbf{Z}^{(i)}\right)\cap An(R_p)$. Since this $S_k$ is conditioned, the path is blocked. Even if there are no such $R_p$ and $R_j$, the path is still blocked by the conditioned $S_k$. If there exists no such $S_k$, then the path contains the bidirected edge between $R_i$ and $W_j$, or the directed path from $W_j$ to $R_i$, which both are contradictions. In conclusion, either (1) there are no such path; or (2) such path is blocked.

Suppose $R_p \rightarrow S_k$. This path is then blocked by conditioning on $R_p$. If there exists no $R_p$ and $R_j$, we can block this path by conditioning on $S_k$, since there should be no bidirected path between $R_i$ and $S_k$. Therefore, either (1) there are no such path; or (2) such path is blocked. This implies that the condition in Eq. (C.10) holds.

We will now prove the **second statement**. We first show

$$P(\mathbf{w}|do(\mathbf{v}\backslash\mathbf{w})) = P(\mathbf{w}|do(\mathbf{v}\backslash\mathbf{s})) = \sum_{\mathbf{s}\backslash\mathbf{w}} P(\mathbf{s}|do(\mathbf{v}\backslash\mathbf{s})). \tag{C.11}$$

Let $\mathbf{W}' \equiv \mathbf{S}\backslash\mathbf{W}$. Then

$$Q[\mathbf{W}] = P(\mathbf{w}|do(\mathbf{v}\backslash\mathbf{w})) = P(\mathbf{w}|do(\mathbf{v}\backslash\mathbf{s}, \mathbf{w}')) \tag{C.12}$$
$$= P(\mathbf{w}|do(\mathbf{v}\backslash\mathbf{s})) \tag{C.13}$$
$$= \sum_{\mathbf{w}'} P(\mathbf{s}|do(\mathbf{v}\backslash\mathbf{s})) \tag{C.14}$$
$$= \sum_{\mathbf{s}\backslash\mathbf{w}} P(\mathbf{s}|do(\mathbf{v}\backslash\mathbf{s})) \tag{C.15}$$

Eq. (C.13) follows by applying Rule 3 of do-calculus using the independence $(\mathbf{W} \perp\!\!\!\perp \mathbf{W}'|\mathbf{V}\backslash\mathbf{S})_{G_{\overline{\mathbf{V}\backslash\mathbf{S}, \mathbf{W}'}}}$. We can show that the independence condition holds using contradiction: Assume there exists a path in $G_{\overline{\mathbf{V}\backslash\mathbf{S}, \mathbf{W}'}}$ between $V_i \in \mathbf{W}$ and $V_j \in \mathbf{W}'$. Such path must have arrows going out of $V_j$, the following node in the path must be in $\mathbf{W}$ for the edge in the path to be in $G_{\overline{\mathbf{V}\backslash\mathbf{S}, \mathbf{W}'}}$. But if this is the case, $V_j$ is a parent of some $V_k \in \mathbf{W}$; then $\mathbf{W}$ is not an ancestral set in $G_{\mathbf{S}}$, a contradiction. This completes the proof that $P(\mathbf{w}|do(\mathbf{v}\backslash\mathbf{w})) = P(\mathbf{w}|do(\mathbf{v}\backslash\mathbf{s})) = \sum_{\mathbf{s}\backslash\mathbf{w}} P(\mathbf{s}|do(\mathbf{v}\backslash\mathbf{s}))$. Note $P(\mathbf{w}|do(\mathbf{v}\backslash\mathbf{s})) = P(\mathbf{w}|do((\mathbf{v}\backslash\mathbf{s}) \cap An(\mathbf{w})))$ by the Rule 3 of do-calculus [5]. This completes the proof. $\square$

### C.1.2 Background Results on Weighted Distributions

**Lemma C.2.** *In Lemma 1, supposing $\mathcal{W}$ satisfies $\mathbb{E}_P[\mathcal{W}(\mathbf{V})] = 1$, then $\mathbb{E}_P[\mathcal{W}(\mathbf{V}) \times \mathcal{W}'(\mathbf{V})] = 1$.*

*Proof.* We first note that $P^{\mathcal{W}}(\mathbf{v})$ is a valid weighted distribution such that $P^{\mathcal{W}}(\mathbf{v}) > 0$ and $\sum_{\mathbf{v}} P^{\mathcal{W}}(\mathbf{v}) = 1$.

Let $\mathbf{X} \equiv (\mathbf{A}\backslash\mathbf{W}) \cap An(\mathbf{W})$. Let $\mathbf{Y} \equiv \mathbf{W}$. Then, $(\mathbf{X}, \mathbf{Y}) = (\mathbf{X} \cup \mathbf{Y}) = \mathbf{A} \cap An(\mathbf{W})$. Let $\mathbf{T} \equiv \mathbf{A}\backslash An(\mathbf{W})$. Then, $(\mathbf{X}, \mathbf{Y}, \mathbf{T}) = \mathbf{X} \cup \mathbf{Y} \cup \mathbf{T} = \mathbf{A}$. Note $\mathcal{W}' \equiv \frac{P^{\mathcal{W}}((\mathbf{a}\backslash\mathbf{w})\cap An(\mathbf{w})|\mathbf{r})}{\prod_{V_i \in (\mathbf{A}\backslash\mathbf{W})\cap An(\mathbf{W})} P^{\mathcal{W}}(v_i|\mathbf{v}^{(i-1)}\cap\mathbf{a}\cap An(\mathbf{w}),\mathbf{r})}$, which is a function of $(\mathbf{R}, \mathbf{X}, \mathbf{Y})$; i.e., $\mathcal{W}' = \mathcal{W}'(\mathbf{R}, \mathbf{X}, \mathbf{Y})$.

Let $\mathcal{W}'' \equiv \mathcal{W} \times \mathcal{W}'$. Then,

$$\mathbb{E}_P\left[\mathcal{W}(\mathbf{V}) \times \mathcal{W}'(\mathbf{V})\right]$$

$$= \mathbb{E}_P\left[\mathcal{W}''(\mathbf{V})\right]$$

$$= \sum_{\mathbf{v}} \mathcal{W}''(\mathbf{v})P(\mathbf{v})$$

$$= \sum_{\mathbf{v}} \mathcal{W}(\mathbf{v})P(\mathbf{v}) \cdot \mathcal{W}'(\mathbf{v})$$

$$= \sum_{\mathbf{a},\mathbf{r}} \underbrace{\left(\frac{\sum_{\mathbf{v}\backslash(\mathbf{a},\mathbf{r})} \mathcal{W}(\mathbf{v})P(\mathbf{v})}{\sum_{\mathbf{v}\backslash\mathbf{r}} \mathcal{W}(\mathbf{v})P(\mathbf{v})}\right)}_{=P^{\mathcal{W}}(\mathbf{a}|\mathbf{r})=Q[\mathbf{A}]} \underbrace{\left(\sum_{\mathbf{v}\backslash\mathbf{r}} \mathcal{W}(\mathbf{v})P(\mathbf{v})\right)}_{=P^{\mathcal{W}}(\mathbf{r})} \cdot \mathcal{W}'(\mathbf{v})$$

$$= \sum_{\mathbf{r}} P^{\mathcal{W}}(\mathbf{r}) \sum_{\mathbf{a}} P^{\mathcal{W}}(\mathbf{a}|\mathbf{r}) \cdot \mathcal{W}'(\mathbf{r},\mathbf{x},\mathbf{y})$$

$$= \sum_{\mathbf{r}} P^{\mathcal{W}}(\mathbf{r}) \sum_{\mathbf{x},\mathbf{y}} \sum_{\mathbf{t}} P^{\mathcal{W}}(\mathbf{x},\mathbf{y},\mathbf{t}|\mathbf{r}) \cdot \mathcal{W}'(\mathbf{r},\mathbf{x},\mathbf{y})$$

$$= \sum_{\mathbf{r}} P^{\mathcal{W}}(\mathbf{r}) \sum_{\mathbf{x},\mathbf{y}} P^{\mathcal{W}}(\mathbf{x},\mathbf{y}|\mathbf{r}) \cdot \mathcal{W}'(\mathbf{r},\mathbf{x},\mathbf{y})$$

$$= \sum_{\mathbf{r}} P^{\mathcal{W}}(\mathbf{r}) \sum_{\mathbf{x},\mathbf{y}} P^{\mathcal{W}}(\mathbf{x},\mathbf{y}|\mathbf{r}) \cdot \frac{P^{\mathcal{W}}(\mathbf{x}|\mathbf{r})}{\prod_{V_i \in \mathbf{X}} P^{\mathcal{W}}\left(v_i|\mathbf{v}^{(i-1)} \cap (\mathbf{x},\mathbf{y}),\mathbf{r}\right)}$$

$$= \sum_{\mathbf{r}} P^{\mathcal{W}}(\mathbf{r}) \sum_{\mathbf{x},\mathbf{y}} \prod_{V_i \in \mathbf{X}} P^{\mathcal{W}}\left(v_i|\mathbf{v}^{(i-1)} \cap (\mathbf{x},\mathbf{y}),\mathbf{r}\right) \cdot \prod_{V_k \in \mathbf{Y}} P^{\mathcal{W}}\left(v_k|\mathbf{v}^{(k-1)} \cap (\mathbf{x},\mathbf{y}),\mathbf{r}\right) \cdot \frac{P^{\mathcal{W}}(\mathbf{x}|\mathbf{r})}{\prod_{V_i \in \mathbf{X}} P^{\mathcal{W}}\left(v_i|\mathbf{v}^{(i-1)} \cap (\mathbf{x},\mathbf{y}),\mathbf{r}\right)}$$

$$= \sum_{\mathbf{r}} P^{\mathcal{W}}(\mathbf{r}) \sum_{\mathbf{x}} P^{\mathcal{W}}(\mathbf{x}|\mathbf{r}) \left(\sum_{\mathbf{y}} \prod_{V_k \in \mathbf{Y}} P^{\mathcal{W}}\left(v_k|\mathbf{v}^{(k-1)} \cap (\mathbf{x},\mathbf{y}),\mathbf{r}\right)\right)$$

$$= 1.$$

where the fourth equality holds by the definition of $Q[\mathbf{A}]$ in Lemma 1 and else equality holds since the $P^{\mathcal{W}}(\mathbf{v})$ is a valid distribution allowing the marginalization; i.e., $\sum_{\mathbf{v}\backslash\mathbf{c}} P^{\mathcal{W}}(\mathbf{v}) = P^{\mathcal{W}}(\mathbf{c})$ for any subset $\mathbf{C} \subseteq \mathbf{V}$, by the definition of the weighted distribution.

$\square$

**Corollary C.1 (Justification of $\mathbb{E}_P[\mathcal{W}] = 1$ for $\mathcal{W}$ in the line 8 of Algo. 1).** *The weight $\mathcal{W}$ in the Line 8 of Algo. 1 satisfies $\mathbb{E}_P[\mathcal{W}(\mathbf{V})] = 1$.*

*Proof.* We first note that $\mathcal{W} = \frac{\prod_{i=1}^{K} P^{\mathcal{W}_{\mathbf{d}_i}}\left(\mathbf{d}_i|\mathbf{r}_{\mathbf{d}_i}\right)}{P(\mathbf{d}|\mathbf{r})} = \frac{\prod_{i=1}^{K} P^{\mathcal{W}_{\mathbf{d}_i}}\left(\mathbf{d}_i|\mathbf{r}_{\mathbf{d}_i}\right)P(\mathbf{v}\backslash\mathbf{d})}{P(\mathbf{v})}$ by the definition of $\mathbf{R}$ in Line 8 of Algo. 1. We note that $P^{\mathcal{W}_{\mathbf{d}_i}}\left(\mathbf{d}_i|\mathbf{r}_{\mathbf{d}_i}\right) = Q[\mathbf{D}_i] = P\left(\mathbf{d}_i|do(\mathbf{v}\backslash\mathbf{d}_i)\right)$. Also, $\prod_{i=1}^{K} P^{\mathcal{W}_{\mathbf{d}_i}}\left(\mathbf{d}_i|\mathbf{r}_{\mathbf{d}_i}\right) = \prod_{i=1}^{K} Q[\mathbf{D}_i] = Q[\mathbf{D}] = P\left(\mathbf{d}|do(\mathbf{v}\backslash\mathbf{d})\right)$. Then,

$$\mathbb{E}_P[\mathcal{W}(\mathbf{V})] = \sum_{\mathbf{v}} \mathcal{W}(\mathbf{v})P(\mathbf{v})$$

$$= \sum_{\mathbf{v}} \prod_{i=1}^{K} P^{\mathcal{W}_{\mathbf{d}_i}}\left(\mathbf{d}_i|\mathbf{r}_{\mathbf{d}_i}\right) P(\mathbf{v}\backslash\mathbf{d})$$

$$= \sum_{\mathbf{v}\backslash\mathbf{d}} P(\mathbf{v}\backslash\mathbf{d}) \sum_{\mathbf{d}} \underbrace{\prod_{i=1}^{K} P^{\mathcal{W}_{\mathbf{d}_i}}\left(\mathbf{d}_i|\mathbf{r}_{\mathbf{d}_i}\right)}_{=P(\mathbf{d}|do(\mathbf{v}\backslash\mathbf{d}))}$$

$$= \sum_{\mathbf{v}\backslash\mathbf{d}} P(\mathbf{v}\backslash\mathbf{d}) \sum_{\mathbf{d}} P\left(\mathbf{d}|do(\mathbf{v}\backslash\mathbf{d})\right) = 1.$$

$\square$

**Lemma C.3** (**Recursion of Weighting**)**.** *Let* $\mathbf{A}, \mathbf{B}$ *be disjoint sets of variables. Let* $\mathbf{C}, \mathbf{D} \subseteq \mathbf{A}$ *be disjoint variables. Let* $q(\mathbf{a}) \equiv P^{\mathcal{W}}(\mathbf{a}|\mathbf{b})$. *Then* $q^{\mathcal{W}'}(\mathbf{c}|\mathbf{d}) = P^{\mathcal{W} \times \mathcal{W}'}(\mathbf{c}|\mathbf{b}, \mathbf{d})$.

*Proof.* We have the following:

$$q^{\mathcal{W}'}(\mathbf{c}|\mathbf{d}) = \frac{\sum_{\mathbf{a}\backslash(\mathbf{c},\mathbf{d})} \mathcal{W}' q(\mathbf{a})}{\sum_{\mathbf{a}\backslash\mathbf{d}} \mathcal{W}' q(\mathbf{a})} = \frac{\sum_{\mathbf{a}\backslash(\mathbf{c},\mathbf{d}))} \mathcal{W}' P^{\mathcal{W}}(\mathbf{a}|\mathbf{b})}{\sum_{\mathbf{a}\backslash\mathbf{d}} \mathcal{W}' P^{\mathcal{W}}(\mathbf{a}|\mathbf{b})} = \frac{\sum_{\mathbf{a}\backslash(\mathbf{c},\mathbf{d})} \mathcal{W}' \frac{P^{\mathcal{W}}(\mathbf{a},\mathbf{b})}{P^{\mathcal{W}}(\mathbf{b})}}{\sum_{\mathbf{a}\backslash\mathbf{d}} \mathcal{W}' \frac{P^{\mathcal{W}}(\mathbf{a},\mathbf{b})}{P^{\mathcal{W}}(\mathbf{b})}},$$

Continuing,

$$\frac{\sum_{\mathbf{a}\backslash(\mathbf{c},\mathbf{d})} \mathcal{W}' \frac{\sum_{\mathbf{v}\backslash(\mathbf{a},\mathbf{b})} \mathcal{W} P(\mathbf{v})}{\sum_{\mathbf{v}\backslash\mathbf{b}} \mathcal{W} P(\mathbf{v})}}{\sum_{\mathbf{a}\backslash\mathbf{d}} \mathcal{W}' \frac{\sum_{\mathbf{v}\backslash(\mathbf{a},\mathbf{b})} \mathcal{W} P(\mathbf{v})}{\sum_{\mathbf{v}\backslash\mathbf{b}} \mathcal{W} P(\mathbf{v})}} = \frac{\frac{\sum_{\mathbf{a}\backslash(\mathbf{c},\mathbf{d}),\mathbf{v}\backslash(\mathbf{a},\mathbf{b})} \mathcal{W}' \times \mathcal{W} \times P(\mathbf{v})}{\sum_{\mathbf{v}\backslash\mathbf{b}} \mathcal{W} P(\mathbf{v})}}{\frac{\sum_{\mathbf{a}\backslash\mathbf{d},\mathbf{v}\backslash(\mathbf{a},\mathbf{b})} \mathcal{W}' \times \mathcal{W} \times P(\mathbf{v})}{\sum_{\mathbf{v}\backslash\mathbf{b}} \mathcal{W} P(\mathbf{v})}}$$

$$= \frac{\sum_{\mathbf{a}\backslash(\mathbf{c},\mathbf{d}),\mathbf{v}\backslash(\mathbf{a},\mathbf{b})} \mathcal{W}' \times \mathcal{W} \times P(\mathbf{v})}{\sum_{\mathbf{a}\backslash\mathbf{d},\mathbf{v}\backslash(\mathbf{a},\mathbf{b})} \mathcal{W}' \times \mathcal{W} \times P(\mathbf{v})}$$

$$= P^{\mathcal{W} \times \mathcal{W}'}(\mathbf{c}|\mathbf{b}, \mathbf{d}).$$

$\square$

**Lemma C.4** (**Marginalization of Weighted Distributions**)**.** *For* $\mathbf{C} \subseteq \mathbf{T}$, $\mathbf{T} \cap \mathbf{X} = \emptyset$, $\sum_{\mathbf{c}} P^{\mathcal{W}}(\mathbf{t}|\mathbf{x}) = P^{\mathcal{W}}(\mathbf{t}\backslash\mathbf{c}|\mathbf{x})$.

*Proof.* We first note $\sum_{\mathbf{c}} P^{\mathcal{W}}(\mathbf{t}, \mathbf{x}) = \sum_{\mathbf{c}} \sum_{\mathbf{v}\backslash(\mathbf{t},\mathbf{x})} P^{\mathcal{W}}(\mathbf{v}) = \sum_{(\mathbf{v}\backslash(\mathbf{t},\mathbf{x}))\cup\mathbf{c}} P^{\mathcal{W}}(\mathbf{v}) = P^{\mathcal{W}}(\mathbf{t}\backslash\mathbf{c}, \mathbf{x})$. Consider the following:

$$\sum_{\mathbf{c}} P^{\mathcal{W}}(\mathbf{t}|\mathbf{x}) = \sum_{\mathbf{c}} \frac{P^{\mathcal{W}}(\mathbf{t}, \mathbf{x})}{P^{\mathcal{W}}(\mathbf{x})} = \frac{\sum_{\mathbf{c}} P^{\mathcal{W}}(\mathbf{t}, \mathbf{x})}{P^{\mathcal{W}}(\mathbf{x})} = \frac{P^{\mathcal{W}}(\mathbf{t}\backslash\mathbf{c}, \mathbf{x})}{P^{\mathcal{W}}(\mathbf{x})} = P^{\mathcal{W}}(\mathbf{t}\backslash\mathbf{c}|\mathbf{x}).$$

$\square$

**Lemma C.5** (**Justification of Line 8 in** `wID`)**.** *For* $\mathbf{D}$ *and* $\mathbf{D}_i$ *(for* $i = 1, 2, \cdots, K$*) in Algo. 1 and* $Q[\mathbf{D}_i] = P^{\mathcal{W}_{\mathbf{d}_i}}(\mathbf{d}_i|\mathbf{r}_{\mathbf{d}_i})$, *let* $\mathcal{W} \equiv (\prod_{i=1}^{K} P^{\mathcal{W}_{\mathbf{d}_i}}(\mathbf{d}_i|\mathbf{r}_{\mathbf{d}_i}))/P(\mathbf{d}|\mathbf{r})$ *where* $\mathbf{R} \equiv \mathbf{V}\backslash\mathbf{D}$. *Then,* $P(\mathbf{y}|do(\mathbf{x})) = P^{\mathcal{W}}(\mathbf{y}|\mathbf{r})$.

*Proof.* We recall that

$$P^{\mathcal{W}}(\mathbf{v}) \equiv \mathcal{W} \cdot P(\mathbf{v}) = \mathcal{W} \cdot P(\mathbf{d}|\mathbf{r})P(\mathbf{r})$$

$$= (\prod_{i=1}^{K} P^{\mathcal{W}_{\mathbf{d}_i}}(\mathbf{d}_i|\mathbf{r}_{\mathbf{d}_i}))/P(\mathbf{d}|\mathbf{r}) \cdot P(\mathbf{d}|\mathbf{r})P(\mathbf{r})$$

$$= P(\mathbf{r}) \prod_{i=1}^{K} P^{\mathcal{W}_{\mathbf{d}_i}}(\mathbf{d}_i|\mathbf{r}_{\mathbf{d}_i}).$$

Also,

$$P^{\mathcal{W}}(\mathbf{r}) = P^{\mathcal{W}}(\mathbf{v}\backslash\mathbf{d}) = \sum_{\mathbf{d}} P^{\mathcal{W}}(\mathbf{v}) = \sum_{\mathbf{d}} P(\mathbf{r}) \prod_{i=1}^{K} P^{\mathcal{W}_{\mathbf{d}_i}}(\mathbf{d}_i|\mathbf{r}_{\mathbf{d}_i}) = P(\mathbf{r}) \sum_{\mathbf{d}} \prod_{i=1}^{K} P^{\mathcal{W}_{\mathbf{d}_i}}(\mathbf{d}_i|\mathbf{r}_{\mathbf{d}_i}) = P(\mathbf{r}).$$

Then,

$$P(\mathbf{y}|do(\mathbf{x})) = \sum_{\mathbf{d}\backslash\mathbf{y}} Q[\mathbf{D}] = \sum_{\mathbf{d}\backslash\mathbf{y}} \prod_{i=1}^{K} P^{\mathcal{W}_{\mathbf{d}_i}}(\mathbf{d}_i|\mathbf{r}_{\mathbf{d}_i}) = \frac{P(\mathbf{v}\backslash\mathbf{d})}{P(\mathbf{v}\backslash\mathbf{d})} \sum_{\mathbf{d}\backslash\mathbf{y}} \prod_{i=1}^{K} P^{\mathcal{W}_{\mathbf{d}_i}}(\mathbf{d}_i|\mathbf{r}_{\mathbf{d}_i})$$

$$= \sum_{\mathbf{d}\backslash\mathbf{y}} \frac{1}{P(\mathbf{v}\backslash\mathbf{d})} P(\mathbf{v}\backslash\mathbf{d}) \prod_{i=1}^{K} P^{\mathcal{W}_{\mathbf{d}_i}}(\mathbf{d}_i|\mathbf{r}_{\mathbf{d}_i}) = \sum_{\mathbf{d}\backslash\mathbf{y}} \frac{1}{P^{\mathcal{W}}(\mathbf{r})} P(\mathbf{v}\backslash\mathbf{d}) \prod_{i=1}^{K} P^{\mathcal{W}_{\mathbf{d}_i}}(\mathbf{d}_i|\mathbf{r}_{\mathbf{d}_i}) = \sum_{\mathbf{d}\backslash\mathbf{y}} \frac{1}{P^{\mathcal{W}}(\mathbf{r})} P^{\mathcal{W}}(\mathbf{v})$$

$$= \sum_{\mathbf{d}\backslash\mathbf{y}} \frac{1}{P^{\mathcal{W}}(\mathbf{r})} P^{\mathcal{W}}(\mathbf{v}) = \sum_{\mathbf{d}\backslash\mathbf{y}} P^{\mathcal{W}}(\mathbf{d}|\mathbf{r}) = P^{\mathcal{W}}(\mathbf{y}|\mathbf{r}).$$

$\square$

## C.2 Proofs

**Lemma C.6** (Restated Lemma 1). *Let a topological order over $\mathbf{V}$ be $V_1 < V_2 < \cdots < V_n$. Suppose $Q[\mathbf{A}]$ is given by $Q[\mathbf{A}] = P^{\mathcal{W}}(\mathbf{a}|\mathbf{r})$ for some $\mathbf{R} \subseteq \mathbf{V}$ and weight function $\mathcal{W}$.*

*1. If $\mathbf{W}$ is a C-component of $G_{\mathbf{A}}$, then $Q[\mathbf{W}] = P^{\mathcal{W} \times \mathcal{W}'}(\mathbf{w}|\mathbf{r}')$, where $\mathbf{R}' \equiv \mathbf{R} \cup ((\mathbf{A}\backslash\mathbf{W}) \cap An(\mathbf{W}))$ and $\mathcal{W}' \equiv \dfrac{P^{\mathcal{W}}((\mathbf{a}\backslash\mathbf{w}) \cap An(\mathbf{w})|\mathbf{r})}{\prod_{V_i \in (\mathbf{A}\backslash\mathbf{W}) \cap An(\mathbf{W})} P^{\mathcal{W}}\left(v_i|\mathbf{v}^{(i-1)} \cap \mathbf{a} \cap An(\mathbf{w}),\mathbf{r}\right)}.$*

*2. If $\mathbf{W} \subseteq \mathbf{A}$ satisfies $\mathbf{W} = An(\mathbf{W})_{G_{\mathbf{A}}}$, then $Q[\mathbf{W}] = P^{\mathcal{W}}(\mathbf{w}|\mathbf{r})$.*

*Proof.* **First statement.** Let $P$ be the joint distribution compatible with $G$. For any subset of nodes $\mathbf{C} \subseteq \mathbf{V}$, let $G(\mathbf{C})$ denote the subgraph of $G$ composing nodes in $\mathbf{C}$. Let $q(\mathbf{a}) \equiv Q[\mathbf{A}] \equiv P(\mathbf{a}|do(\mathbf{v}\backslash\mathbf{a})) = P^{\mathcal{W}}(\mathbf{a}|\mathbf{r})$ denote a joint distribution over $\mathbf{A}$. We note that $q(\mathbf{a})$ is a valid distribution, since $\sum_{\mathbf{a}} q(\mathbf{a}) = 1$ and $q(\mathbf{a}) \geq 0$. Since $q(\mathbf{a}) \equiv P(\mathbf{a}|do(\mathbf{v}\backslash\mathbf{a}))$, $G_{\overline{\mathbf{V}\backslash\mathbf{A}}}(\mathbf{A})$ is a graph compatible with $q(\mathbf{a})$. For any nodes $\mathbf{B}, \mathbf{C} \subseteq \mathbf{A}$, we will note that $q(\mathbf{b}|do(\mathbf{c}))$ denote the distribution over $\mathbf{B}$ induced by not only fixing $\mathbf{V}\backslash\mathbf{A} = \mathbf{v}\backslash\mathbf{a}$ in $G$ (which induced $q(\mathbf{a})$), but also fixing $\mathbf{C} = \mathbf{c}$ in $G$. That is, $q(\mathbf{b}|do(\mathbf{c})) = P(\mathbf{b}|do(\mathbf{v}\backslash\mathbf{a},\mathbf{c}))$.

Let $\mathbf{W}$ be a C-component of $G_{\mathbf{A}}$ (i.e., $G(\mathbf{A})$). We note that this $\mathbf{W}$ is also a C-component of $G_{\overline{\mathbf{V}\backslash\mathbf{A}}}(\mathbf{A})$ since no edges between nodes in $\mathbf{A}$ are cut. Now, consider $Q[\mathbf{W}] \equiv P(\mathbf{w}|do(\mathbf{v}\backslash\mathbf{w}))$. We note the following equality holds:

$$Q[\mathbf{W}] \equiv P(\mathbf{w}|do(\mathbf{v}\backslash\mathbf{w})) = P(\mathbf{w}|do(\mathbf{v}\backslash\mathbf{a},\mathbf{a}\backslash\mathbf{w})) = q(\mathbf{w}|do(\mathbf{a}\backslash\mathbf{w})) = q(\mathbf{w}|do((\mathbf{a}\backslash\mathbf{w}) \cap An(\mathbf{w}))).$$

The equality $P(\mathbf{w}|do(\mathbf{v}\backslash\mathbf{a},\mathbf{a}\backslash\mathbf{w})) = q(\mathbf{w}|do(\mathbf{a}\backslash\mathbf{w}))$ holds by the above discussion about the definition of $q(\cdot)$. The equality $q(\mathbf{w}|do(\mathbf{a}\backslash\mathbf{w})) = q(\mathbf{w}|do((\mathbf{a}\backslash\mathbf{w}) \cap An(\mathbf{w})))$ holds since

$$\begin{aligned} q(\mathbf{w}|do(\mathbf{a}\backslash\mathbf{w})) &= P(\mathbf{w}|do(\mathbf{a}\backslash\mathbf{w},\mathbf{v}\backslash\mathbf{a})) \\ &= P(\mathbf{w}|do((An(\mathbf{w}) \cap \mathbf{a}\backslash\mathbf{w}),\mathbf{v}\backslash\mathbf{a})) \\ &= q(\mathbf{w}|do((\mathbf{a}\backslash\mathbf{w}) \cap An(\mathbf{w}))), \end{aligned}$$

where the third equality holds by the above discussion about the definition of $q(\cdot)$. The second equality holds by

$$(\mathbf{W} \perp\!\!\!\perp (\mathbf{A}\backslash\mathbf{W})\backslash An(\mathbf{W})|An(\mathbf{W}) \cap (\mathbf{A}\backslash\mathbf{W}),\mathbf{V}\backslash\mathbf{A})_{G_{\overline{\mathbf{A}\backslash\mathbf{W},\mathbf{V}\backslash\mathbf{A}}}}.$$

Specifically, in $G_{\overline{\mathbf{A}\backslash\mathbf{W},\mathbf{V}\backslash\mathbf{A}}}$, for $W_k \in \mathbf{W}$ and $A_j \in (\mathbf{A}\backslash\mathbf{W})\backslash An(\mathbf{W})$, the only possible path between $W_k$ and $A_j$ is the path from $A_j$ to $W_k$. However, such path is contradictory since $A_j$ is not an ancestor of $W_k$. Then, by Rule 3 of $do$-Calculus, the second equality holds.

We note that, in $G_{\overline{\mathbf{V}\backslash\mathbf{A}}}(\mathbf{A})$ (where the distribution $q(\mathbf{a})$ is compatible with), $\emptyset$ satisfies mSBD criterion relative to $((\mathbf{A}\backslash\mathbf{W}) \cap An(\mathbf{W}), \mathbf{W})$ by Lemma C.1. This means that, for the $q(\mathbf{a})$, the interventional distribution $q(\mathbf{w}|do(\mathbf{a}\backslash\mathbf{w} \cap An(\mathbf{w})))$ is given by the mSBD adjustment. Specifically, since since $\emptyset$ satisfies mSBD criterion relative to $(\mathbf{A}\backslash\mathbf{W} \cap An(\mathbf{W}), \mathbf{W})$ in $G_{\overline{\mathbf{V}\backslash\mathbf{A}}}(\mathbf{A})$ (where the graph $G_{\overline{\mathbf{V}\backslash\mathbf{A}}}$ induces the joint distribution $q(\mathbf{a})$), by Thm. C.2, $q(\mathbf{w}|do(\mathbf{a}\backslash\mathbf{w} \cap An(\mathbf{w}))) = q^{\mathcal{W}'}(\mathbf{w}|(\mathbf{a}\backslash\mathbf{w}) \cap An(\mathbf{w}))$ where $\mathcal{W}' \equiv \dfrac{q((\mathbf{a}\backslash\mathbf{w}) \cap An(\mathbf{w}))}{\prod_{V_i \in (\mathbf{A}\backslash\mathbf{W}) \cap An(\mathbf{W})} q(v_i|\mathbf{v}^{(i-1)} \cap \mathbf{A} \cap An(\mathbf{w}))}$. Then, by Lemma C.3, given the fact that $q(\mathbf{a}) = P^{\mathcal{W}}(\mathbf{a}|\mathbf{r})$,

$$q^{\mathcal{W}'}(\mathbf{w}|(\mathbf{a}\backslash\mathbf{w}) \cap An(\mathbf{w})) = P^{\mathcal{W} \times \mathcal{W}'}(\mathbf{w}|(\mathbf{a}\backslash\mathbf{w}) \cap An(\mathbf{w}),\mathbf{r}),$$

where, by Lemma C.3,

$$\begin{aligned} \mathcal{W}' &\equiv \frac{q((\mathbf{a}\backslash\mathbf{w}) \cap An(\mathbf{w}))}{\prod_{V_i \in (\mathbf{A}\backslash\mathbf{W}) \cap An(\mathbf{W})} q(v_i|\mathbf{v}^{(i-1)} \cap \mathbf{a} \cap An(\mathbf{w}))} \\ &= \frac{P^{\mathcal{W}}((\mathbf{a}\backslash\mathbf{w}) \cap An(\mathbf{w})|\mathbf{r})}{\prod_{V_i \in (\mathbf{A}\backslash\mathbf{W}) \cap An(\mathbf{W})} P^{\mathcal{W}}\left(v_i|\mathbf{v}^{(i-1)} \cap \mathbf{a} \cap An(\mathbf{w}),\mathbf{r}\right)}. \end{aligned}$$

This completes the proof.

**Second statement.** Under the given condition, $Q[\mathbf{W}] = \sum_{\mathbf{a}\backslash\mathbf{w}} Q[\mathbf{A}]$ by [8, Lemma 3]. Therefore, $Q[\mathbf{W}] = \sum_{\mathbf{a}\backslash\mathbf{w}} P^{\mathcal{W}}(\mathbf{a}|\mathbf{r}) = P^{\mathcal{W}}(\mathbf{w}|\mathbf{r})$.

$\square$

**Theorem C.3** (**Restated Theorem 1**). *A causal effect $P(\mathbf{y}|do(\mathbf{x}))$ is identifiable if and only if $\mathtt{wID}(\mathbf{x},\mathbf{y},G,P)$ (Algo. 1) returns $P^{\mathcal{W}}(\mathbf{y}|\mathbf{r})$ such that $P(\mathbf{y}|do(\mathbf{x})) = P^{\mathcal{W}}(\mathbf{y}|\mathbf{r})$.*

*Proof.* Algo. 1 follows precisely Tian's algorithm (Alg. 2 in [8]) for identifying causal effects except that in Lines 3, 9, a.1, and a.6 the Q-factors are expressed in the form of weighted distributions. The correctness of Lines 3, a.1, and a.6 follows from Lemma 1. The correctness of Line 9 follows from Lemma C.5. Then the soundness and completeness of Algo. 1 follows from the soundness and completeness of Tian's algorithm [2].

$\square$

**Theorem C.4** (**Restated Theorem 2**). *Let $h^* \equiv \arg\min_{h \in \mathcal{H}} \mathcal{R}^{\mathcal{W}^*}(h)$, and $(\mathcal{W}_m, h_m) \equiv \arg\min_{\mathcal{W} \in \mathcal{H}_{\mathcal{W}}, h \in \mathcal{H}} \mathcal{L}(\mathcal{W}, h)$, where $\mathcal{H}_{\mathcal{W}}$ is the model hypotheses class for $\mathcal{W}$. Suppose $\mathcal{H}_{\mathcal{W}}$ is correctly specified such that $\mathcal{W}^* \in \mathcal{H}_{\mathcal{W}}$. Then, $h_m$ converges to $h^*$ with a rate of $O_p(m^{-1/4})$. Specifically, $\mathcal{R}^{\mathcal{W}^*}(h_m) - \mathcal{R}^{\mathcal{W}^*}(h^*) \leq O_p(m^{-1/4})$.*

*Proof.* We rewrite the objective function as follow:

$$\mathcal{L}(\mathcal{W}, h)$$

$$\equiv \widehat{\mathcal{R}}^{\mathcal{W}}(h) + \frac{\lambda_h}{m}C(h) + \sqrt{\frac{1}{m}\sum_{i=1}^{m}\left(\mathcal{W}(\mathbf{V}_{(i)}) - \mathcal{W}^*(\mathbf{V}_{(i)})\right)^2 + \frac{\lambda_{\mathcal{W}}}{m}\|\mathcal{W}\|_2}$$

$$= \widehat{\mathcal{R}}^{\mathcal{W}}(h) + \underbrace{O_p(m^{-1})}_{=(\lambda_h/m)C(h)} + \sqrt{\mathbb{E}_P\left[\left(\mathcal{W}(\mathbf{V}_{(i)}) - \mathcal{W}^*(\mathbf{V}_{(i)})\right)^2\right]} + O_p(m^{-1/4}) + O_p(m^{-1/2})$$

$$= R^{\mathcal{W}}(h) + \sqrt{\mathbb{E}_P\left[\left(\mathcal{W}(\mathbf{V}_{(i)}) - \mathcal{W}^*(\mathbf{V}_{(i)})\right)^2\right]} + O_p(m^{-1/4}).$$

To see the above equality, let $A_m \equiv \frac{1}{m}\sum_{i=1}^{m}\left(\mathcal{W}(\mathbf{V}_{(i)}) - \mathcal{W}^*(\mathbf{V}_{(i)})\right)^2$ and $\mu \equiv \mathbb{E}_P\left[\left(\mathcal{W}(\mathbf{V}_{(i)}) - \mathcal{W}^*(\mathbf{V}_{(i)})\right)^2\right]$. Then,

$$P(\sqrt{m} \cdot |A_m - \mu| \geq t) \leq 2 \cdot \exp\left(-\frac{2t^2}{c^2}\right),$$

implying that $A_m - \mu = O_P(m^{-1/2})$. Then, $\sqrt{A_m} = \sqrt{\mu + O_P(m^{-1/2})} = \sqrt{\mu} + O_P(m^{-1/4})$. Also, since $\frac{\lambda_{\mathcal{W}}}{m}\|\mathcal{W}\|_2 = O_P(m^{-1})$, $\sqrt{\frac{\lambda_{\mathcal{W}}}{m}\|\mathcal{W}\|_2} = O_P(m^{-1/2})$. This implies that $\mathcal{L}(\mathcal{W}^*, h) = \mathcal{R}^{\mathcal{W}^*}(h) + O_p(m^{-1/4})$.

Now, consider Prop. 1 with respect to $m$. Since $\log(m) \leq m^{-1/4}$ for $m \leq 10000$, we note

$$F(p, m, \delta) = O\left((\log(m)/m)^{3/8}\right) \leq O\left(m^{3/32}/m^{-3/8}\right) = O_P(m^{-9/32}).$$

Then, $m^{1/4}F(p, m, \delta) = O_P(m^{-1/32}) = O_P(1)$, implying that $F(p, m, \delta) = O_P(m^{-1/4})$. Therefore, we can rewrite Prop. 1 with respect O $m$ as $\mathcal{R}^{\mathcal{W}^*}(h) \leq \widehat{\mathcal{R}}^{\mathcal{W}}(h) + \mathbb{E}_P\left[|\mathcal{W}^* - \mathcal{W}|\right] + O_p(m^{-1/4})$. Then,

$$\mathcal{R}^{\mathcal{W}^*}(h_m) - \mathcal{R}^{\mathcal{W}^*}(h^*)$$

$$\leq \widehat{\mathcal{R}}^{\mathcal{W}_m}(h_m) + \mathbb{E}_P\left[(\mathcal{W}^* - \mathcal{W}_m)\right] + O_p(m^{-1/4}) - \mathcal{R}^{\mathcal{W}^*}(h^*)$$

$$\leq \underbrace{\widehat{\mathcal{R}}^{\mathcal{W}_m}(h_m) + \sqrt{\mathbb{E}_P\left[(\mathcal{W}^* - \mathcal{W}_m)^2\right]} + O_p(m^{-1/4})}_{=\mathcal{L}(\mathcal{W}_m, h_m)} + \mathbb{E}\left[|\mathcal{W}^* - \mathcal{W}_m|\right] - \sqrt{\mathbb{E}_P\left[(\mathcal{W}^* - \mathcal{W}_m)^2\right]} + O_p(m^{-1/4}) - \mathcal{R}^{\mathcal{W}^*}(h^*)$$

$$= \mathcal{L}(\mathcal{W}_m, h_m) + \underbrace{\mathbb{E}\left[|\mathcal{W}^* - \mathcal{W}_m|\right] - \sqrt{\mathbb{E}_P\left[(\mathcal{W}^* - \mathcal{W}_m)^2\right]}}_{\leq 0 \text{ By Hoelder's inequality}} + O_p(m^{-1/4}) - \underbrace{\mathcal{R}^{\mathcal{W}^*}(h^*)}_{=\mathcal{L}(\mathcal{W}^*, h^*) + O_p(m^{-1/4})}$$

$$\leq \underbrace{\mathcal{L}(\mathcal{W}_m, h_m) - \mathcal{L}(\mathcal{W}^*, h^*)}_{\leq 0 \text{ by definition of } (h_m, \mathcal{W}_m)} + O_p(m^{-1/4}) + O_p(m^{-1/4}).$$

$$\leq O_p(m^{-1/4}).$$

This completes the proof.

□

**Theorem C.5** (**Restated Theorem 3**)**.** *Let $m = |\mathcal{D}|$ and $n \equiv |\mathbf{V}|$. Assume all weights satisfy $0 < \mathcal{W} < c$ for some constant $c > 0$. Let $T_1(m)$ denote the time complexity for estimating $\widehat{P}(v_i|\cdot)$ from sample $\mathcal{D} \sim P(\mathbf{v})$ for $V_i \in \mathbf{V}$. Let $K$ denote the number of C-factors in $G_{\mathbf{D}}$ (in Algo. 1). Let $T_2(m)$ denote the time complexity of minimizing $\mathcal{L}_{\mathcal{W}}$ and $\mathcal{L}_h$. Then, Algo. 2 runs in $O\left(poly(n) + nK(mc + nT_1(m)) + T_2(m)\right)$ time, where $O\left(poly(n)\right)$ is for running Algo. 1, $O\left(nK(mc + nT_1(m))\right)$ for evaluating $\widehat{\mathcal{W}^*}$.*

*Proof.* Algo. 1 is a precise replication of the identification algorithm in [8] which is known to have time complexity $O\left(poly(n)\right)$. That evaluating $\widehat{\mathcal{W}^*}$ takes $O\left(nK(mc + nT_1(m))\right)$ is proved in Lemma B.4. Time complexities to optimize the loss functions $\mathcal{L}_{\mathcal{W}}, \mathcal{L}_h$ are $T_2(m)$. This completes the proof.

□

# D    Further Details in Experiments

**Tuning hyperparameters.** Throughout the experiments, the hyperparameters $\lambda_{\mathcal{W}}, \lambda_h$ in Eq. (6) are chosen using the grid-search method [6]. Specifically, the hyperparameter $\lambda_{\mathcal{W}}$ is chosen as follows: (1) Split the sample as $\mathcal{D} = \mathcal{D}_{tr} \cup \mathcal{D}_{te}$ at random; (2) For each fixed $\lambda_k \in \{2, 4, \cdots, 50\}$, learn $\mathcal{W}_k \equiv \arg\min_{\mathcal{W}'} \mathcal{L}_{\mathcal{W}}(\mathcal{W}', \lambda_k; \widehat{\mathcal{W}^*})$ from $\mathcal{D}_{tr}$ and compute $\epsilon_{k,te} \equiv \mathcal{L}_{\mathcal{W}}(\mathcal{W}_k, \lambda_k; \widehat{\mathcal{W}^*})$ on $\mathcal{D}_{te}$; and (3) Choose $k' \equiv \arg\min_k \{\epsilon_{k,te}\}_{k \in \{2,4,\cdots,50\}}$ and set $\lambda_{\mathcal{W}} \equiv \lambda_{k'}$. With the fixed learned $\mathcal{W}$, we choose $\lambda_h$ analogously.

## D.1    Structural Causal Models Used in the Experiments

**Example 1.** A data generating process written in R is given in the following:

```
varval = 2

c1 = rnorm(D,1,1)
c2 = rnorm(D,-2,1)
cz = rnorm(D,2,1)

U1mean = -8; U1Var = 10
U1 = rnorm(N,U1mean,U1Var)
U1.intv = rnorm(Nintv,U1mean,U1Var)

U2mean = 6; U2Var = 8
U2 = rnorm(N,U2mean,U2Var)
U2.intv = rnorm(Nintv,U2mean,U2Var)

fW = function(N,U1,U2){
  Uw = rnorm(N,0,0.5)
  W = matrix(0,ncol=D,nrow=N)
  for (idx in 1:D){
    W[,idx] = rbinom(N,size=1,prob=inv.logit(c1[idx]*U1+c2[idx]*U2))
  }
  W = data.frame(W)
  colnames(W) = paste('W',1:D,sep="")
  return(W)
}
fZ = function(N,W){
  Uz = rnorm(N,0,0.5)
  Wmat = as.matrix(2*W-1)
  czmat = as.matrix(cz)
  Zval = inv.logit(Wmat %*% czmat)
  Z = round(inv.logit(-1*Zval + Uz-1 ))
```

```
      return (Z)
}
fX = function (N,U1,Z){
   Ux = rnorm (N,1 ,6)
   X = rbinom (N, size =1, inv . logit (1*U1 − 2*Z + Ux − 5   ))
      return (X)
}
fY = function (N,U2,X){
   Uy = rnorm (N,−2 ,1)
   ind .X = 2*X − 1
   Y = rbinom (N, size =1, inv . logit (0.5*U2 − 2*ind .X + Uy))
      return (Y)
}
```

**Example 2.** A data generating process written in R is given in the following:

```
varval = 1

c1 = rnorm (D,−2 ,0.5)
c2 = rnorm (D,1 ,0.5)

cx = rnorm (D,2 ,0.5)
cz = rnorm (D,−0.8 ,0.5)
cy = rnorm (D,1.5 ,0.5)

U1 = rnorm (N,0 , varval )
U2 = rnorm (N,0 , varval )
U3 = rnorm (N,0 , varval )
U1intv = rnorm (Nintv ,0 , varval )
U2intv = rnorm (Nintv ,0 , varval )
U3intv = rnorm (Nintv ,0 , varval )

fW = function (N,U1,U2){
   Uw = rnorm (N,0 ,0.5)
   W = matrix (0 , ncol=D, nrow=N)
   for (idx in 1:D){
      W[ , idx ] = rbinom (N, size =1, prob=inv . logit (c1 [ idx ]*U1+c2 [ idx ]*U2 +Uw))
   }
   W = data . frame (W)
   colnames (W) = paste ( 'W' ,1:D, sep="")
      return (W)
}
fX = function (N,W,U1,U3){
   Ux = rnorm (N,0 ,0.5)
   Wmat = as . matrix (2*W−1)
   cxmat = as . matrix (cx )
   Wval = inv . logit (Wmat %*% cxmat )
   X = rbinom (N, size =1, inv . logit (−1*Wval − 2*U1 + 0.5*U3*Wval + Ux − 2*U1*U3 ))
      return (X)
}
fZ = function (N,W,X){
   Uz = rnorm (N,0 ,1)
   Wmat = as . matrix (2*W−1)
   czmat = as . matrix (cz )
   Wval = inv . logit (Wmat %*% czmat )
   Z = rbinom (N, size =1, inv . logit (1*Wval − 2*(2*X−1) + Uz ))
      return (Z)
}
fY = function (N,U2,U3,Z,W){
   Uy = rnorm (N,0 ,0.5)
   Wval = myXOR(W)
   Y = rbinom (N, size =1, inv . logit (−U3−U2+Z−10*Wval +1))
      return (Y)
}
```

**Example 3.** A data generating process written in R is given in the following:

```r
c.z.1 = rnorm(D,-2,0.5); c.z.2 = rnorm(D,1,0.5); c.z.3 = rnorm(D,0,1)
c.w.1 = rnorm(D,2,0.5); c.w.2 = rnorm(D,-1,0.5) ; c.w.3 = rnorm(D,1,0.5)
cx = rnorm(D,2,0.5); cr = rnorm(D,-1,1); cz = rnorm(D,-2,0.3)

U1 = rnorm(N,-1,varval); U2 = rnorm(N,-0.5,varval);
U3 = rnorm(N,0.5,varval); U4 = rnorm(N,1,varval)

fW = function(N,U1,U2){
  Uw = rnorm(N,0,0.5)
  W = matrix(0,ncol=D,nrow=N)
  for (idx in 1:D){
    W[,idx] = rbinom(N,size=1,prob=inv.logit(c.w.1[idx]*U1+c.w.2[idx]*U2 + Uw))
  }
  W = data.frame(W)
  colnames(W) = paste('W',1:D,sep="")
  return(W)
}

fX = function(N,W,U1,U3){
  Ux = rnorm(N,0,0.5)
  Wmat = as.matrix(2*W-1)
  cxmat = as.matrix(cx)
  Wval = inv.logit(Wmat %*% cxmat)
  X = rbinom(N,size=1,inv.logit(-1*Wval + -0.5*U1 - 0.2*U3 + Ux-2 ))
  return(X)
}

fR = function(N,W,U4){
  Ur = rnorm(N,0,0.5)
  Wmat = as.matrix(2*W-1)
  crmat = as.matrix(cr)
  Wval = inv.logit(Wmat %*% crmat)
  R = rbinom(N,size=1,inv.logit(-1*Wval - 1.2*U4 + Ur - 2))
  return(R)
}

fZ = function(N,W,X,R,U4){
  Uz = rnorm(N,0,0.5)
  Wmat = as.matrix(2*W-1)
  czmat = as.matrix(cz)
  Wval = inv.logit(Wmat %*% czmat)
  Z = rbinom(N,size=1,inv.logit(0.5*Wval+U4 + 0.5*(2*X-1) -
  0.9*(2*R-1) + Uz-1 - log(abs(Wval)+1) ))
  return(Z)
}

fY = function(N,R,Z,U2,U3){
  Uy = rnorm(N,0,0.5)
  Y = rbinom(N,size=1,inv.logit(-1*(2*R-1)*Z +
  0.5*(2*Z-1)*log(abs(U2*U3)+1) -
  R*U2- Uy +1))
  return(Y)
}
```

## D.2 Additional Experimental Results

In this section, we provide experimental results of evaluating the proposed WERM based estimators against Plug-in in Examples 1, 2, and 3 for $D \equiv |W| \in \{5, 10\}$.

**Example 1 (Fig. 1b).** We test on estimating $\mathbb{E}[Y|do(x)]$ with $D \in \{5, 10\}$ where the causal effect $P(y|do(x))$ is given by Eq. (A.1). The MAAE plots are given in Fig. (D.1a,D.1d). We observe that the WERM-based methods (WERM-ID/WERM-ID-R) significantly outperform Plug-in.

(a) Example 1 (Fig. 1b)    (b) Example 2 (Fig. 2a)    (c) Example 3 (Fig. 2b)

(d) Example 1 (Fig. 1b)    (e) Example 2 (Fig. 2a)    (f) Example 3 (Fig. 2b)

Figure D.1: **(Top)** MAAE plots comparing proposed WERM based estimators (WERM-ID and WERM-ID-R) with Plug-in on $D = 5$. **(Bottom)** Plots on $D = 10$.

**Example 2 (Fig. 2a).** We test on estimating $\mathbb{E}\left[Y|do(x)\right]$ with $D \in \{5, 10\}$ where the effect $P\left(y|do(x)\right)$ is given by Eq. (A.2). The MAAE plots are given in Fig. (D.1b,D.1e) We observe that the WERM-based methods (WERM-ID/WERM-ID-R) perform on par with Plug-in.

**Example 3 (Fig. 2b).** We test on estimating $\mathbb{E}\left[Y|do(x,r)\right]$ with $D \in \{5, 10\}$ where $P\left(y|do(x,r)\right)$ is given by Eq. (A.3). The MAAE plots are given in Fig. (D.1c,D.1f). We note that WERM-ID-R significantly outperforms WERM-ID, and both significantly outperform Plug-in.

### D.3 Comparison with potential outcome frameworks (For Reviewer 3)

(a) Example 1 (Fig. 1b)    (b) Example 2 (Fig. 2a)    (c) Example 3 (Fig. 2b)

Figure D.2: **(For Reviewer 3)** MAAE plots comparing the proposed vs. potential outcome based estimator for Example (1,2,3) with $D = 15$. Shades are standard deviations.

In this section, we compare the proposed estimator with the potential-outcome (PO) based estimator (specifically, the inverse probability weighting estimator) to address the question of Reviewer 3: "*I am a bit curious about the comparison results with some recent causal inference methods under PO framework if simply seeing the whole other variables* $\mathbf{V}\backslash\{\mathbf{X}, \mathbf{Y}\}$ *as observed confounders.*" Comparison examples are given in Fig. (D.2a,D.2b,D.2c). As expected, the performances of the PO framework based estimator are inferior to the proposed estimator ('WERM-ID-R'). This result implies adjusting covariates without taking into account the causal graph might yield inaccurate estimates of the causal effect.