[Reviews · NeurIPS 2020]

Review 1

Summary and Contributions: The paper transforms the identifiable causal inference problem as a weighted empirical risk minimization problem. It proposes a new objective for WERM and a learning algorithm based on it. The simulation shows WERM-ID achieves low absolute error and computational time.

Strengths: The paper provides a general framework to cast the causal estimand to a form of weighted ERM. The method is widely applicable to a variety of causal graphs. This paper contains rich materials. Viewing causal inference from an optimization perspective is relevant to the NeurIPS community.

Weaknesses: 1. Under the identification assumption, if other causal functionals exist, such as regression adjustment and IPTW, what is the advantage of the WERM-ID? 2. In the empirical simulation, there proposed method is only compared with plug-in estimands. What about other weighting methods? 3. The paper contains many technical details but lacks a detailed explanation. If the page limit is a major concern, the author may need to decide what are the essential messages to express and explain them very clearly, instead of thrusting and skipping over many materials. See the Clarity section for more details. ###### post rebuttal comment: The author's feedback addresses the major concerns of the reviewer. I modified the evaluation accordingly to marginal above acceptance.

Correctness: The paper claims the algorithm can “generate ANY identifiable causal functionals as weighted distributions”. This claim excludes any counterexample; is it too strong? Is the identification as the only assumption that is needed for WERM?

Clarity: The explaining example is not very clear. Some formulations need further explanation: i) Why is Eq. (1) true? ii) What does the dash curve with double arrow in Figure 1 mean? It is not a standard notation in DAG. Is there a cycle X -> W ->R ->X? iii) Why p(y | do(x)) = p(y | do(r), x)? iv) There are also many skipped steps in the derivation above Lemma 1 for P(y|do(x)) = P (x, y|do(r)) /P (x|do(r)) = P(y|do(r), x) = P^W(y|x, r)

Relation to Prior Work: The connection with previous weighting methods is not well explained.

Reproducibility: Yes

Additional Feedback:


Review 2

Summary and Contributions: The paper tackles the problem of assessing the strength of the causal effect in the presence of unobserved common causes from observational data when the assumed causal relationships among the observed variables are given in the form of a "causal" graph. In this, the paper extends the work of "On the identification of causal effects" [48] by formulating the problem of learning functionals as a weighted ERM problem.

Strengths: A polynomial-time algorithm that is able to learn from finite samples that improve the performance over the proposed baseline. Elaborate derivations and presumably correct proofs (I did not check the proofs).

Weaknesses: The paper is, somewhat surprisingly, is extremely difficult to follow. It is written in some form of nonlinear fashion with thoughts and references jumping, at times, back across a few pages. Although based on [48], this paper is much harder to follow unlike it, and is lacking the context of what and why is being done. In this sense, it is hardly self-contained. Potentially, the jargon and context may be clear to narrow specialists working precisely in this topic, but I doubt a wider machine learning readership, even well versed in causal learning, will find the paper easy to follow. I did invest substantial time and only after going through [48] some things became clearer, but not all. The lack of clarity could have been attributed to the page limit and the needed to be dense. However, the introduction practically occupies 3 pages out of 8 and doesn't explain that the causal graph needs to be provided beforehand and we only need this new method because of unobserved common causes, since otherwise the causal graph and P(V) completely define everything. In general, the paper feels like a good fit for a journal, where it can be safely combined with the supplemented proofs and additional experiments on more realistic graphs larger than the 3-4 variable examples could be given. The intended readership could be statisticians and practical relevance to any graphs of interesting scale is irrelevant, I leave that as a possibility. The example graphs are unrealistic for interesting practical applications in ML. The polynomial-time algorithm runs in close to a few seconds if not faster, and yet no empirical evidence of performance is given on large graphs. Potentially, sample complexity may be exponential in the node number and the algorithm can only handle 3-4 nodes. This is not clear from the paper.

Correctness: I did not check the proofs in the supplement. Empirical validation is extremely limited as explained above.

Clarity: Either written for a very narrow audience or is just poorly written, overwhelming rather than explaining.

Relation to Prior Work: Relevant literature is discussed

Reproducibility: No

Additional Feedback:


Review 3

Summary and Contributions: In this paper, the authors provide an approach for estimating causal effects by weighted predictors. It is a good try to causal estimation. Surprisingly, it is revealed that causal effects representation can be converted to weighted distributions.

Strengths: The paper provides solid theory about converting the representation of identified causal effects to the format of weighted distribution. Moreover, under the assumption that W*\in \mathcal{H}_W, the authors prove the consistency and converge rate of the estimated causal effect.

Weaknesses: I guess it is a bit not friendly to readers who are not quite clear about this region.

Correctness: They seem right. But I did not check them throughly.

Clarity: Yes. But it is a bit difficult to read.

Relation to Prior Work: Yes.

Reproducibility: Yes

Additional Feedback: I am a bit curious about the comparison results with some recent causal inference methods under PO framework if simply seeing the whole other variables (V\{X,Y}) as observed confounders. I know that PO framework does not take the causal graph into account and it is not reasonable from the viewpoint of identifiability in this setting. I am just curious about the performance gap between the proposed method and the SOTA in related regions. Because although the method in this paper is theoretically reasonable, more errors may be introduced due to the complicated computation (such as estimating many conditional possibilities). So I have this question. By the comparison, at least it can tell which present method for estimating causal effect is better for a user without much related knowledge but just wants to estimate the causal effect. I totally believe that the framework in this paper has unlimited development possibilities and could have better and better performance in the long run. This paper is a good start. So the comparison results will not influence my judgment on this paper. Supplementary Page 279: Should "Thm. 1" be "Prop. 1"?


Review 4

Summary and Contributions: This paper provides a learning framework for causal inference when the conditional ignorability assumption does not hold although the causal effect is identifiable. The framework provides an algorithm to first massage the desired estimand (using weighted distributions) to look like a back-door expression (on which we can apply the conventional Empirical Risk Minimization (ERM) technique) and then learn the required weights on the previous step to complete the training procedure. The algorithm works in an end-to-end fashion.

Strengths: Parts of the claimed contributions of this work are significant and to the best of my knowledge, has not been explored before. Causal inference is a relevant subject to the NeurIPS community.

Weaknesses: I have many questions regarding the soundness of the claims; especially theoretical grounding. I have elaborated on them in the “correctness” section.

Correctness: Eq. (1): What happened to the variable “r”? Lines 89-90: Why does “P(x, y, w | do(r))” equal “1/P(r | w) * P(x, y, w, r)”? Algorithm 1, line 8.5: What is “T”? Learning low variance weights is not novel as (Swaminathan and Joachims, 2015) have already addressed it in their Counterfactual Risk Minimization (CRM) framework. Please comment on whether/how your work differs from theirs? Swaminathan, A., & Joachims, T. (2015). Counterfactual risk minimization: Learning from logged bandit feedback. In International Conference on Machine Learning (ICML).

Clarity: Use of too many inline mathematical statements (some even essential to understanding an idea) has made it difficult to keep the reading flow.

Relation to Prior Work: There is a nice selection of literature reviewed in the paper; however, there is no explicit discussion on how this work differs from the prior work.

Reproducibility: No

Additional Feedback: Please address my comments in the rebuttal and I shall update my score accordingly. ===== post-rebuttal comments ===== I appreciate the authors' response to my questions. Although these cover my concerns, I think they should incorporate them into the final manuscript. I have updated my score to 6.

[Author Response · NeurIPS 2020]

We thank the reviewers for their time and valuable feedback. We believe that a few misreadings of our results may have
led to a somewhat negative evaluation, which we ask for reconsideration given the clarifications provided below.
**Connection with previous works and contributions.** Under the structural causal model framework (Pearl, 2000), there
has been extensive work on the problem of causal effect identification (as cited in line 30), which determines whether
the causal functional could be obtained from observed distribution given a causal graph (i.e., *identifiable*) and derives
such functional whenever identifiable. One outstanding challenge to applying these identification results in practical
settings is that there has been no sample and computationally efficient estimators working for *any* identifiable causal
functional. This paper addresses this challenge by filling the gap between causal "*identification*" and "*estimation*". We
develop a learning framework that could work for *any* identifiable causal functional beyond the ignorability assumption.
**Clarity.** The paper aims to fill the gap from causal effect identification to estimation and assumes a basic background
in identification theory. The discussion regarding Eq. (1) in line 85-94 and 140-151 is to show that it's possible to
*manually* convert a functional output by a standard identification algorithm, but not friendly for the WERM framework,
into a weighted form using known identification techniques. Non-familiarity with the identification techniques will not
impact the rest of the paper, as this work develops a systematic algorithm to achieve the task (ref. line 181-184).
**Reply to Reviewer 1.** *"if other causal functionals exist, such as regression adjustment and IPTW, what is the advantage*
*of the WERM-ID?"* Good question. The only setting where regression-based and weighting based estimators both
exist is when the ignorability assumption holds (e.g., Fig. 1(a)). In this case, the proposed WERM estimator reduces
to the standard re-weighting based estimators, which one can estimate using any ML methods (cited in line 58).
*"...only compared with plug-in estimands. What about other weighting methods?"* As noted in lines 39 and 319, the
plug-in estimator is the only viable estimator known to date for arbitrary identifiable functionals. Other weighting
methods are not applicable as mentioned in lines (39-40, 320-321). *"This claim excludes any counterexample; is it*
*too strong?"* This claim is a major contribution of this paper. We show *any* identifiable causal functionals can be
converted in the weighted distribution format, and estimated using the WERM framework (Thm. 1,2). *"What does*
*the dash curve with double arrow in Figure 1 mean?"* As noted in line 112 and following the convention (Pearl,
2000), the dashed-bidirected arrows between $(X, Y)$ encode unobserved causes between $(X, Y)$; i.e., $X \leftarrow U \rightarrow Y$,
where $U$ is unobserved. *"Why is Eq. (1) true?"* As stated in line 79, one could derive Eq. (1) by running a standard
identification algorithm (e.g., [48] or [45]). *"There are also many skipped steps in the derivation above Lemma 1 for*
$P(y|do(x)) = P(x, y|do(r))/P(x|do(r)) = P(y|do(r), x) = P^W(y|x, r)$*"* The first equality is due to Eq. (1), and
explanation in line 140-146; the 2nd is definition of conditional probability; the last is from the equation in line 148.
*"The connection with previous weighting methods..."* Existing weighting estimators were developed for settings where
the ignorability assumption holds. Our work proposes a novel method working for *any identifiable* causal functionals.
**Reply to Reviewer 2.** *"doesn't explain that the causal graph needs to be provided beforehand"* We respectfully
disagree since we explicitly state that the identification problem assumes a given causal graph in line 25-28, and in the
subsequent example line 31-35. *"no empirical evidence of performance is given on large graphs... the algorithm can*
*only handle 3-4 nodes"* We respectfully note that neither the theorems nor the empirical simulations limit the proposed
algorithm to small graphs. The time complexity is *polynomial* in the size of the graph (Thm. 3) and empirically
demonstrated in Fig. 3(d,e,f). In the experiments, the covariates $W$ is set to be a vector of $D$ binary variables (line 302),
with $D = 15$ in Fig. 3(a,b,c) (stated in line 304, 307, 310).
**Reply to Reviewer 3.** *"I am a bit curious ..."* Great suggestion. A
comparison example is given in Fig. 1. As expected, the performance
of the PO framework based estimator is inferior to the proposed
estimator ('WERM-ID-R'). This result implies adjusting covariates
without taking into account the causal graph might yield inaccurate
estimates of the causal effect; we'll add this to the paper, thanks.

**Figure 1: (For Reviewer 3)** A MAAE plot comparing the proposed vs. PO-based estimator for Example 3 ($D = 15$). Shades are standard deviations.

**Reply to Reviewer 7.** *"Eq. (1): What happened to the variable r?"*
That the r.h.s of (1) is independent of the value $r$ is known as a Verma
constraint on the observed distribution implied by the causal graph.
This issue is discussed in Appendix A line 61-68. *"Lines 89-90: Why*
*does $P(x, y, w|do(r))$ equal $(1/P(r|w))P(x, y, w, r)$?"* This can be derived by a standard identification algorithm
(e.g., in [48] or [45]), or directly using Theorem 1 in [48]. *Algorithm 1, line 8.5: What is "T"?* $\mathbf{T}$ is an arbitrary set.
Procedure wIdentify($\mathbf{C}, \mathbf{T}, Q[\mathbf{T}], \mathbf{r}, \mathcal{W}$) outputs $Q[\mathbf{C}]$ for $\mathbf{C} \subseteq \mathbf{T}$ given input $\mathbf{T}$ and $Q[\mathbf{T}] = P^{\mathcal{W}}(\mathbf{t}|\mathbf{r})$. *"Learning*
*low variance weights is not novel as (Swaminathan and Joachims, 2015) have already addressed..."* We appreciated the
great work of SJ15 [47] and cited it in line (58, 200, 216). We adopted the idea in SJ15 of learning low variance weights.
However, the results (which deals with ignorable cases / propensity score weighting) are not directly applicable, and
properties such as learning guarantee in Thm. 2 needs to be re-derived in our context. *"there is no explicit discussion on*
*how this work differs from the prior work"* The prior work on applying WERM to causal inference is limited to settings
contingent on the ignorability/backdoor condition (line 56-61). This work developed a general learning framework that
fully brings together the causal identification theory and WERM methods (as summarized in line 95-106).

[Meta-Review · NeurIPS 2020]

The paper applies weighted ERM to learn the causal functionals given the graph. The proposed framework converts the causal effect into weighted distribution without conditional ignorability assumption. We agree that learning causal effect without ignorability is very interesting and could lead to further advances in causal inference. But, the clarity of the theoretical analysis and the experimental results received much criticism during the discussions. Overall, the theoretical contributions are strong, but the paper needs a lot of work to improve its clarity. Also, the material from the rebuttal should be incorporated into the main paper to strengthen the motivation and the connections with prior work.